# TRPV1 Hyperfunction Contributes to Renal Inflammation in Oxalate Nephropathy

**DOI:** 10.3390/ijms22126204

**Published:** 2021-06-08

**Authors:** Chien-Lin Lu, Te-Yi Teng, Min-Tser Liao, Ming-Chieh Ma

**Affiliations:** 1Division of Nephrology, Department of Medicine, Fu Jen Catholic University Hospital, New Taipei 24352, Taiwan; janlin0123@gmail.com; 2School of Medicine, Fu Jen Catholic University, New Taipei 242062, Taiwan; 3Department of General Dentistry, Taoyuan Armed Forces General Hospital, Taoyuan 32551, Taiwan; teyiteng@gmail.com; 4Department of Pediatrics, Taoyuan Armed Forces General Hospital, Taoyuan 32551, Taiwan; liaoped804h@yahoo.com.tw

**Keywords:** TRPV1, arachidonate 12-lipoxygenase, 12(*S*)-HETE, calcium, oxidative stress, inflammasome, oxalate, hyperoxaluria, inflammation

## Abstract

Inflammation worsens oxalate nephropathy by exacerbating tubular damage. The transient receptor potential vanilloid 1 (TRPV1) channel is present in kidney and has a polymodal sensing ability. Here, we tested whether TRPV1 plays a role in hyperoxaluria-induced renal inflammation. In TRPV1-expressed proximal tubular cells LLC-PK_1_, oxalate could induce cell damage in a time- and dose-dependent manner; this was associated with increased arachidonate 12-lipoxygenase (ALOX12) expression and synthesis of endovanilloid 12(*S*)-hydroxyeicosatetraenoic acid for TRPV1 activation. Inhibition of ALOX12 or TRPV1 attenuated oxalate-mediated cell damage. We further showed that increases in intracellular Ca^2+^ and protein kinase C α activation are downstream of TRPV1 for NADPH oxidase 4 upregulation and reactive oxygen species formation. These trigger tubular cell inflammation via increased NLR family pyrin domain-containing 3 expression, caspase-1 activation, and interleukin (IL)-1β release, and were alleviated by TRPV1 inhibition. Male hyperoxaluric rats demonstrated urinary supersaturation, tubular damage, and oxidative stress in a time-dependent manner. Chronic TRPV1 inhibition did not affect hyperoxaluria and urinary supersaturation, but markedly reduced tubular damage and calcium oxalate crystal deposition by lowering oxidative stress and inflammatory signaling. Taking all these results together, we conclude that TRPV1 hyperfunction contributes to oxalate-induced renal inflammation. Blunting TRPV1 function attenuates hyperoxaluric nephropathy.

## 1. Introduction

The transient receptor potential vanilloid 1 (TRPV1) channel is known as a polymodal sensor recognizing the presence of heat (>43 °C), acidosis (pH ≦ 5.9), and irritating chemicals such as lipid metabolites, capsaicinoids, and oxygen radicals [1,2]. In the sensory nervous system, TRPV1 integrates these noxious stimuli for induction of pain sensation. Aside from nociception, we previously showed a novel role of TRPV1 in the visceral sensation of the rat kidney, where it acts as a mechanoreceptor [3]. TRPV1 is located at the renal pelvic wall and expressed in the peripheral nerve endings of renal sensory nerves to trigger mechano-sensation during urine excretion. Activation of TRPV1-mediated renal sensory nerves reflexly reduces efferent renal sympathetic nerve activity and enhances urinary excretion in terms of the diuretic renorenal reflex [3]. Meanwhile, we also showed that TRPV1 is expressed in the renal tubules, with strong expression in the apical membrane [3]. The tubular expression of TRPV1 in the rat kidney was consistent with previous findings [4]. However, the novel role of TRPV1 in renal tubules is still unclear. With the property of sensing noxious compounds, we postulate that TRPV1 in renal tubules may detect noxious chemicals such as oxalate after glomerular filtration and coordinate tubular function for clearance of these compounds. Oxalate is removed from body principally by the kidneys [5]. Studies have demonstrated that during renal excretion, oxalate is toxic to renal tubular cells not only at supraphysiological (hyperoxaluric) concentration, but also at physiological concentration in human urine [6,7]. Preventing oxalate from encountering renal tubular cells, such as by supplementing with probiotics with oxalate-degrading ability in the intestines, effectively attenuates hyperoxaluria-mediated tubular damage effects [8,9]. Once tubular cells are damaged by oxalate, cell debris can shed into supersaturated urine and accelerate calcium oxalate (CaOx) crystal formation by acting as seeds for nephrolithiasis.

We previously showed that TRPV1 hyperfunction contributes to uremic toxin indoxyl sulfate (IS)-induced renal tubular damage in a cell model of chronic kidney disease [10]. IS increased arachidonate 12-lipoxygenase (ALOX12) expression and endogenous ligand 12(*S*)-hydroxyeicosatetraenoic acid (12(*S*)-HETE) to activate TRPV1 in renal tubular cell damage [10]. Interestingly, lipid metabolites such as arachidonic acid and eicosanoids produced by lipoxygenase are known to provoke an inflammatory response [10]. Since inflammation plays an important role in the disease progression of nephrolithiasis [11,12], we hypothesized that the 12(*S*)-HETE/TRPV1 pathway would play a role in hyperoxaluria-induced tubular damage and CaOx formation. This hypothesis is supported by previous findings showing that enhanced TRPV1 expression in the epithelial cells of urethra, bladder, gastrointestinal tract, and lung are usually associated with tissue inflammation and lead to functional defects in these organs [13]. This indicates that TRPV1 connects between noxious sensing and inflammation under pathophysiological conditions. One of the ways the disease progression manifests from hyperoxaluria to CaOx nephropathy to end-stage renal failure is through inflammation-induced tubular damage [12,14,15]. A variety of stimuli such as crystals, Ca^2+^ load, and reactive oxygen species (ROS) production in oxidative stress have been demonstrated to activate inflammasome NLR family pyrin domain-containing (NLRP) 3, which, in turn, triggers inflammatory cytokine release [16,17,18]. CaOx crystal itself can induce inflammasome-mediated tissue injury, necroptosis signaling, and receptor-interacting protein kinase-3 in triggering tubular cell necrosis [19]. Moreover, previous studies showed that a sustained rise in intracellular Ca^2+^ concentration ([Ca^2+^]i) via store-operated Ca^2+^ entry mechanism in renal tubular cells after CaOx crystal internalization can induce endoplasmic reticular stress, ROS production, and cell death [20]. Interestingly, increased [Ca^2+^]i and oxidative stress via TRPV1 have been identified to play a role in the regulation of cell death and survival [21]. Based on this information, we tested the possible role of Ca^2+^-permeable TRPV1 in oxalate-mediated tubular cell damage and whether this relies on events of inflammation in both in vitro and in vivo models.

## 2. Results

### 2.1. Blockade of TRPV1 Attenuates Oxalate-Induced Cytotoxicity in Renal Tubular Cells

Oxalate is known to be toxic to tubular cells [7]. Our results confirmed this by showing that oxalate increased lactate dehydrogenase (LDH) release, a reliable cell damage marker for necrosis, in both a dose- and time-dependent manner (Figure 1A). We further demonstrated that TRPV1 is present in LLC-PK_1_ cells with an abundant distribution in plasma membrane (Figure 1B). Oxalate treatment did not affect TRPV1 expression, but significantly increased ALOX12 expression in a time-dependent manner. A slight but insignificant low TRPV1 level was found after 24 h of oxalate treatment (Figure 1C). Interestingly, there was a parallel increase in the cellular content of 12(S)-HETE in the oxalate-treated cells compared to the control cells (Figure 1D). Moreover, inhibition of ALOX12 by a selective blocker, cinnamyl-3,4-dihydroxy-α-cyanocinnamate (CDC), and TRPV1 by two specific blockers, capsazepine (Capz) and SB-366791 (SB), attenuated the tubulotoxicity of oxalate by lowering LDH release (Figure 1E). These results indicate that oxalate-induced tubular cell damage is dependent on the effects of ALOX12 and TRPV1. Increased 12(*S*)-HETE formation may overactivate TRPV1, because it is known as an endogenous ligand for TRPV1.

### 2.2. Oxalate-Induced Intracellular Ca^2+^ Level ([Ca^2+^]_i_) Elevation and PKC Activation

As TRPV1 is a Ca^2+^-permeable channel, we then examined whether changes in [Ca^2+^]i caused by TRPV1 participate in the effect of oxalate. Compared to the control cells, oxalate increased [Ca^2+^]i at the time point of 24 h (Figure 2A). Inhibition of TRPV1 by SB or treatment of Ca^2+^ chelation by1,2-bis-(2-aminophenoxy)ethane-N,N,N′,N′-tetraacetic acid tetra(acetoxymethyl) ester (BAPTA) in the oxalate-treated tubular cells significantly lowered the increase in [Ca^2+^]i caused by oxalate. SB or BAPTA alone showed no effect on [Ca^2+^]i.

Since PKCα is an important signaling molecule in kidney disease [22], we investigated whether it plays a downstream role in Ca^2+^-permeable TRPV1 after oxalate treatment. Comparing protein translocation from the cytosol to the plasma membrane, cytosolic PKCα expression was significantly lower, while its membranous expression was higher in the oxalate-treated cells compared to control cells (Figure 2B). Calculating the ratio of membranous to cytosolic protein level to represent changes in protein activity, PKCα activity was found to be significantly higher in the oxalate-treated cells than in the control cells (Figure 2B). This indicates that oxalate activates PKCα for signal pathways in tubular cell damage. Inhibition of TRPV1 and Ca^2+^ chelation reversed PKCα translocation caused by oxalate, indicating that PKCα activation is downstream of TRPV1.

### 2.3. Nox4 but Not Nox2 Is Activated by PKC

The enzymatic activity of Nox is the major source of intracellular ROS generation, including superoxide and H_2_O_2_. We previously showed that oxidative stress caused by excess ROS formation plays an important role in tubular damage in kidney stone patients with hyperoxaluria [23]. The present results confirm tubular cell expression of Nox2 and Nox4 (Figure 3A,B). Interestingly, oxalate increased Nox4 but not Nox2 expression at both the mRNA and protein level. Moreover, inhibition of TRPV1 (SB) and PKC (Gö 6976, Go) significantly attenuated Nox4 upregulation in the oxalate-treated cells (Figure 3A,B), indicating TRPV1-mediated PKCα activation is the underlying mechanism for Nox4 upregulation. We examined whether excess ROS formation is related to increased Nox4 expression in terms of oxidative stress. Oxalate treatment increased the release of superoxide and H_2_O_2_ into culture medium (Figure 3C). Inhibition of TRPV1 (SB) and Nox4 (GKT137831, GKT) significantly attenuated the effect of oxalate on ROS generation.

### 2.4. Blockade of ALOX12/TRPV1/Nox4 Pathway Attenuates Inflammatory Response

Since inflammation plays a pivotal role in disease progression in oxalate nephrolithiasis [12,24], we then tested whether changes in inflammasome NLRP3 expression could be found in oxalate-treated cells. Our results show that oxalate significantly increased NLRP3 expression, and this could be attenuated by co-treatment with TRPV1 blocker (SB), Ca^2+^ chelator (BAPTA), and Nox4 inhibitor (GKT) (Figure 4A). This clearly indicates that oxalate-induced Ca^2+^ overload and oxidative stress are mediators for triggering inflammation in tubular cells.

Activation of NLRP3 induces caspase-1 activation and IL-1β release in oxalate-mediated inflammatory injury [15]. Therefore, we investigated whether there are parallel changes in caspase-1 activity and IL-1β release with NLRP3 expression. Our results show that TRPV1 inhibition, Ca^2+^ chelation, and Nox4 inhibition significantly attenuated the increased caspase-1 activity caused by oxalate (Figure 4B). These blockers also significantly attenuated the cleavage of IL-1β precursor (pro-IL-1β at 31 kDa) to the mature form of IL-1β (at 17 kDa) in oxalate-treated cells (representative blots in Figure 4C). We then calculated the ratio of protein expression between the mature form of IL-1β and pro-IL-1β to represent the degree of IL-1β activation. Oxalate treatment significantly increased IL-1β activation, and this could be attenuated by TRPV1 inhibition, Ca^2+^ chelation, or Nox4 inhibition (bar graph in Figure 4C). Oxalate and these blockers alone, however, showed little effect on pro-IL-1β expression. To further validate the role of IL-1β in response to oxalate, we quantitated the amount of IL-1β secretion into culture medium and showed that IL-1β release is significantly increased by oxalate (Figure 4D). This was markedly attenuated by TRPV1 blockade, Ca^2+^ chelation, or Nox4 inhibition.

### 2.5. Hyperoxaluria Upregulates ALOX12 and TRPV1 in Rat Kidney

To confirm the in vitro observations of tubular inflammation, we then investigated the pro-inflammatory role of the ALOX12/TRPV1 pathway in the in vivo rat model of hyperoxaluria. Hyperoxaluria was successfully induced in rats fed with the hydroxyl-L-proline (HP) diet in a time-dependent manner from 3 to 28 days (Figure 5A). Interestingly, this was associated with increased urinary Kidney Injury Molecule 1 (KIM-1) excretion, a specific marker of proximal tubule damage (Figure 5B). A peak increase in urinary KIM-1 excretion was found after 14 days of induction. 

Compared to the control group, TRPV1 expression was increased on the 7th day in HP kidneys and gradually elevated after 14 and 28 days compared to control kidneys (Figure 5C). This was associated with increased expression of ALOX12 after 3 days of induction. ALOX12 levels remained high thereafter in the HP kidneys. 

In control kidneys, immunohistochemical staining revealed that TRPV1 expression was present in renal tubules, with strong expression at the apical membrane (upper pictures in Figure 5D). Tubular expression of TRPV1 was significantly increased on the 14th day in HP kidneys, not only at the apical membrane but also inside tubular cells. ALOX12 was abundantly expressed in the renal tubules of control kidneys (lower pictures in Figure 5D). ALOX12 was also expressed in renal vessels but less distributed in the glomerulus. On the 14th day in HP kidneys, tubular and vascular expression of ALOX12 was largely increased compared to control kidneys. ALOX12 expression also appeared in the glomerulus of HP kidney.

Interestingly, increased renal expression of ALOX12 and TRPV1 was associated with increased renal content of 12(S)-HETE in a time-dependent manner, with a peak increase on the 7th day in HP kidney compared to control kidney (Figure 5E).

### 2.6. Hyperoxaluria Induces Oxidative Stress and Renal Inflammation

We previously showed that both Nox2 and Nox4 are present in rat kidney [25]. This study shows that Nox2 in renal cortex was increased after 7 days of hyperoxaluric induction only (Figure 6A). However, Nox4 expression was increased in HP kidneys after 3 days of induction, and this increase persisted to 28 days (Figure 6B).

Using in vivo chemiluminescence (CL) recordings, our results show that intrarenal arterial infusion of lucigenin into the HP kidney induced a gradual rise in CL counts on the kidney surface (representative tracing in Figure 6C). The CL counts always fluctuated around baseline in control kidney. The AUC results show that lucigenin-dependent CL counts were markedly increased in HP kidneys in an induction time-dependent manner compared with control kidneys (left bar graph in Figure 6C).

Inflammation plays a pivotal role in the functional deterioration of hyperoxaluric kidneys [24], and here we show that NLRP3 expression abruptly increased after 3 days of induction and persisted thereafter in HP kidneys (Figure 6D). More downstream of NLRP3, we found that active form (cleavage) of caspase-1 at 20 kDa was increased in HP kidneys. This was associated with a gradual decline in the expression of pro-caspase-1 at 45 kDa (representative blots in Figure 6E). Compared to control kidney, caspase-1 activity was increased in HP kidney in a time-dependent manner, with a peak increase on day 14, by calculating the ratio of protein expression between cleavage of caspase-1 and pro-caspase-1 (lower bar graph in Figure 6E). Increased caspase-1 activity in HP kidneys was associated with increased renal content of IL-1β, with a peak increase on day 14 (Figure 6F).

### 2.7. Blockade of TRPV1 Ameliorates Tubular Damage and CaOx Formation by Hyperoxaluria

Because there were parallel changes in TRPV1, ALOX12, and NLRP3 expression as well as caspase-1 activation and IL-1β formation in HP kidneys, we examined whether inhibition of TRPV1 would ameliorate hyperoxaluria-induced renal inflammation and tubular injury. Chronic delivery of two specific TRPV1 blockers, Capz and SB, for 7 and 14 days via subcutaneous mini-osmotic pump significantly attenuated hyperoxaluria-induced renal hypertrophy (as the ratio of kidney to body weight) (Figure 7). This also was associated with reductions in CaOx crystal formation, kidney injury (as tubular damage marker KIM-1), and the weight of urine sediments. Blockade of TRPV1 showed no effect on the degree of hyperoxaluria or urinary supersaturation.

### 2.8. TRPV1 Inhibition Attenuates Oxidative Stress and Renal Inflammation

In the 14 day treated kidneys, TRPV1 inhibition by Capz or SB significantly reduced Nox4 expression caused by hyperoxaluria (Figure 8A). This was associated with decreased lucigenin-dependent CL counts in HP kidneys (Figure 8B). This indicates that Nox4-mediated oxidative stress is involved in hyperoxaluria-induced kidney injury. Moreover, SB was more effective than Capz at attenuating Nox4-mediated ROS formation in HP kidneys. 

We then investigated the pro-inflammatory role of TRPV1 in the 14 day HP kidneys. Hyperoxaluria-induced renal NLRP3 upregulation and increased caspase-1 activity and renal content of IL-1β in HP kidneys were reduced by TRPV1 inhibition (Figure 8C–E). However, Capz or SB alone showed no effect on these parameters.

## 3. Discussion

As the schematic diagram in Figure 9 shows, oxalate activates ALOX12 after uptake by tubular cells and enhances endovanilloid 12(S)-HETE formation for TRPV1 activation. Increased [Ca^2+^]i via TRPV1 induces PKCα activation by increasing PKCα translocation, which subsequently upregulates Nox4 to liberate superoxide and H_2_O_2_. Oxidative stress and Ca^2+^ trigger renal tubular inflammation by increased NLRP3 expression and caspase-1 activation to increase IL-1β formation and release in oxalate-mediated tubular cell injury. TRPV1 hyperfunction in oxalate-mediated nephropathy is further evidenced by in vivo observations showing that chronic TRPV1 inhibition markedly reduced hyperoxaluria-induced tubular damage and CaOx crystal formation via inflammation. However, TRPV1 hyperfunction showed no effect on the degree of hyperoxaluria or urinary supersaturation.

Our results are consistent with previous findings showing that oxalate is toxic to tubular cells by increasing LDH release, a marker for plasma membrane damage in necrosis [6,7,26]. Concerning the dose of oxalate, a previous study showed that oxalate in the proximal tubule has a concentration around 0.002–0.1 mM and increases to 0.1–0.5 mM in the collecting ducts [6]. Oxalate in the range of 0.1–0.5 mM is also the concentration for daily excretion in normal human urine [7,27]. This indicates that distal tubular cells are exposed to oxalate at concentrations 5- to 50-fold higher than proximal tubular cells. Interestingly, a study by Schepers et al. [6] showed that treatment with 0.5 mM oxalate for 24 h did not affect LDH release in Madin-Darby canine kidney (MDCK) cells, which is a cell line usually used to represent the origin of distal tubular cells. However, a study by Thamilselvan et al. [7] showed that treatment with 0.55 mM oxalate for 24 h significantly increased LDH release in LLC-PK_1_ cells, the same cell line used in this study to represent the origin of proximal tubular cells. These results indicate that proximal tubular cells are more vulnerable to oxalate than distal tubular cells. The dose of 0.5 mM oxalate used in this study showed results consistent with induced LDH release in LLC-PK_1_ cells in a time-dependent manner (Figure 1A). Indeed, the 0.5 mM oxalate used in this study mimics the oxalate concentration in human urine when excess oxalate from food is excreted [7,27]. Our animal model also confirmed this, because oxalate concentration in urine was around 0.52–0.87 mM after hyperoxaluric induction.

As in our previous observations [10], TRPV1 was present in LLC-PK_1_ cells (Figure 1B,C). Moreover, blockade of TRPV1 significantly attenuated oxalate-induced LDH release (Figure 1E), indicating that TRPV1 is involved in oxalate-mediated tubular cell damage. However, this was not due to changes in protein expression of TRPV1, because it was not affected by the oxalate in tubular cells (Figure 1C). In the hyperoxaluric rats, we found that TRPV1 expression was upregulated after 7 days of induction (Figure 5C). The discrepancy between in vitro and in vivo observations can be explained by different treatment or induction time. TRPV1 upregulation in the rat kidney was mostly located at the luminal side of the renal tubules (Figure 5D), and this possibly allows the renal tubule to detect the presence of excess noxious stimuli in ultrafiltrate such as oxalate after secretion by the proximal tubule. One could hypothesize that the function of Ca^2+^-permeable TRPV1 may actually be protective against crystallization since it’s an apically-localized Ca^2+^ importer (Figure 5D), which in the presence of oxalate, would be upregulated to remove Ca^2+^ from the lumen to prevent its complex with oxalate. Here we however showed a pro-inflammatory role of TRPV1 in tubular damage. In peripheral blood mononuclear cells from patients with end-stage kidney disease, TRPV1 upregulation increased susceptibility to endovanilloid N-arachidonoyl-dopamine-induced cell death [28]. Our results are therefore comparable to previous findings demonstrating that cells overexpressing TRPV1 are more susceptible to vanilloid-induced cell death [1,29,30,31,32]. Moreover, TRPV1 hyperfunction was related to increased ALOX12 expression (Figure 1C,D), which is responsible for the synthesis of endovanilloid 12(*S*)-HETE via its lipoxygenase activity [31]. The in vitro and in vivo results are consistent in showing increased expression of ALOX12 and amount of 12(*S*)-HETE, indicating the importance of the ALOX12/12(*S*)-HETE pathway in oxalate or hyperoxaluric nephropathy. Inhibition of ALOX12 by a selective blocker, CDC, confirmed that this pathway plays a role in inducing TRPV1 hyperfunction in oxalate-induced tubular damage (Figure 1E). However, how oxalate upregulates ALOX12 is unclear. A previous study revealed that three upstream genes (runt-related transcription factor 1, tumor protein p63, and phospholipid hydroperoxide glutathione peroxidase) regulate the protein expression of ALOX12 [33]. Further study is required to examine whether oxalate affects the transcriptional activity of these factors on ALOX12.

Inappropriate enhanced function of ALOX12, 12(*S*)-HETE, and TRPV1 is known to trigger an inflammatory response that exacerbates organ damage [34,35,36]. Previous studies have demonstrated that renal inflammation plays a pivotal role in functional loss in oxalate nephropathy [24]. Interestingly, inhibition of TRPV1 attenuates oxalate- or hyperoxaluria-induced NLRP3 upregulation (Figure 4 and Figure 7). NLRP3 is an intracellular sensor that is responsible for detecting a range of exogenous and endogenous danger signals for subsequent formation and activation [15]. Two signals, the “priming” signal induced by Toll-like receptor activation and the “activating” signal triggered by ion flux, ROS generation, or lysosomal rupture, participate in oxalate-induced activation of NLRP3 [11,24].

Here we show that abnormal ion influx and ROS generation in the “activating” signal are, at least, involved in TRPV1-mediated NLRP3 upregulation. The in vitro results in this study show that oxalate increases [Ca^2+^]i via TRPV1 (Figure 2A), and this probably promotes the assembly of inflammasome components [16]. Although we did not directly test whether Ca^2+^ chelation affects NLRP3 activation, blockade of Ca^2+^-permeable TRPV1 attenuates NLRP3 upregulation (Figure 4A). Previous studies demonstrated that oxalate can induce increases in [Ca^2+^]i not only in the renal tubular cells, but also in other cells such as neuroblastoma, endothelial, and rat dorsal root ganglia (DRG) cells [37,38,39,40]. In oxalate-treated rat DRG cells, increased [Ca^2+^]i can be attenuated by L-type Ca^2+^ channel blockers [40]. Here we show a similar result, but the most increased [Ca^2+^]i caused by oxalate was dependent on TRPV1 (Figure 2A). The remaining increased [Ca^2+^]i not abolished by TRPV1 inhibition was possibly due to the effect of L-type Ca^2+^ channels, because it was also present in LLC-PK_1_ cells [41].

The other trigger for NLRP3 activation in oxalate-induced tubular damage is ROS [42]. Here we show that oxalate increases O_2_^−^ and H_2_O_2_ formation, attenuated by TRPV1 inhibition (Figure 3C). To consider how oxalate increases ROS, we found that Nox4 more than Nox2 is the source for ROS generation (Figure 3A,B and Figure 6A,B). This is consistent with our previous findings showing that protein expression of gp91phox (a main subunit of Nox2) was unaffected in the renal cortex of hyperoxaluric kidney after long-term induction (42 days) [43]. Renal Nox2 expression was only increased in 7 day hyperoxaluric kidneys, and this was also comparable to our previous findings [44]. Moreover, TRPV1 inhibition in the tubular cells and the 14 day HP kidneys also did not affect Nox2 expression in this study. These results suggest a minor role of Nox2 in hyperoxaluric nephropathy. 

Nox4 (also termed renal oxidase or Renox) was originally identified in the renal cortex because of abundant expression [25]. The catalytic activity of Nox4 is homologous to Nox2 for ROS generation; unlike Nox2, Nox4 activation does not require p47phox, p67phox, or the small GTPase Rac [25]. Our previous study showed that H_2_O_2_ generated by Nox4 can activate TRPV1 in the sensory terminals of afferent renal nerves located at the renal pelvis [25]. Therefore, we speculate that increased ROS formation by Nox4 may enhance the function of tubular TRPV1 in terms of renal inflammation. 

Inhibition of PKCα in tubular cells by Gӧ 6976 largely abrogates Nox4 upregulation caused by oxalate (Figure 3B), indicating that PKCα acts upstream for regulation of Nox4 expression. PKCα inhibition also reduces ROS formation caused by oxalate (Figure 3C). This result is consistent with the previous finding of Thamilselvan et al. [45] showing that PKC activation is responsible for Nox upregulation in oxalate-mediated oxidative injury. Considering that ROS activates inflammasome, the NLRP3-thioredoxin-interacting protein (TXNIP) has been demonstrated to act as a sensor of the changing levels ROS derived from Nox and dissociated from the antioxidant thioredoxin [46]. TXNIP then binds with NLRP3 and leads to NLRP3 formation and activation. 

Although we did not test the direct effect of ROS scavenge on the effect of NLRP3, Nox4 inhibition significantly attenuates NLRP3 upregulation and subsequent caspase-1 activation caused by oxalate (Figure 4A). Upon activation by Ca^2+^ or ROS, NLRP3 assembles and activates caspase-1, which subsequently cleaves pro-IL-1β for maturation to IL-1β and triggers innate immunity in oxalate/CaOx nephropathy [11,47]. NLRP3 is not only present in the phagocytes but is also expressed in the renal tubular cells of both rat and human kidneys [15]. Our observations confirmed this, showing the presence of NLRP3 in tubular cells and rat kidney (Figure 4 and Figure 8). Tubular distribution of NLRP3 could be affected by oxalate treatment or hyperoxaluric induction via TRPV1 and increased caspase-1 activation for inflammatory cytokine IL-1β formation (Figure 4 and Figure 8).

TRPV1 is widely known to play a pro-inflammatory role, but studies also point to its anti-inflammatory and protective roles in kidney diseases [2,48,49]. Different ways of TRPV1 activation or inhibition and the multiple functional presence of distinct subcellular distributions of TRPV1 might explain the discrepancy of observations [50]. Previous studies reported an association between TRPV1 activation and inflammation-induced epithelial cell damage in tissues other than kidney. TRPV1-mediated Ca^2+^ influx and enhanced arachidonic acid metabolism play a role in capsaicin-induced respiratory epithelial cell death and release of pro-inflammatory cytokines IL-6 and IL-8 [51]. In corneal epithelial cells, TPRV1 activation increases IL-6 and IL-8 expression and is associated with corneal epithelial cell injury after alkali burn [52]. In human esophageal epithelial cells, TRPV1 activation by excess acid induces the release of inflammatory mediators such as IL-8, and this can be blocked by TRPV1 antagonists [53]. The present results are consistent with these findings, as the TRPV1 hyperfunction is associated with activation of inflammatory signals triggering IL-1β release in hyperoxaluria-induced nephropathy.

In conclusion, the present study suggests that TRPV1 hyperfunction caused by overstimulation of 12(*S*)-HETE or by channel upregulation contributes to renal inflammation in oxalate nephropathy. Our results demonstrate that the TRPV1-mediated Ca^2+^/PKCα/Nox4 pathway is important for ROS production and subsequent NLRP3/caspase-1 activation for increased IL-1β release in renal tubular cells. The pro-inflammatory role of TRPV1 hyperfunction was further confirmed in hyperoxaluric rat kidney. Blockade of TRPV1 mitigates the detrimental effect of inflammation-induced tubular damage in both in vitro and in vivo models of hyperoxaluria, indicating a potential role of TRPV1 antagonism in the treatment of oxalate nephropathy.

## 4. Materials and Methods

### 4.1. Tubular Cell Culture and Drug Treatment

TRPV1-expressed Lilly Laboratories Cell-Porcine Kidney 1 (LLC-PK_1_, pig renal tubular epithelial kidney cells) were used as an in vitro model of proximal tubule cell origin, as previously described [10]. Cells were purchased from the Bioresource Collection and Research Center (Hsinchu, Taiwan). This cell line was originally derived from the American Type Culture Collection line CL-101. All culture media and supplements were purchased from Thermo Scientific HyClone (South Logan, UT, USA). Cells were cultured at 37 °C/5% CO_2_ in Medium 199 containing 3% fetal bovine serum, sodium bicarbonate (1.5 g/L), penicillin (10,000 U/mL), and streptomycin (10,000 μg/mL). Cells were maintained and subcultured every 3 days when they reached confluence. Cells were seeded into a 6-well plate for 2 days. On the day of the experiment, 100 μL of culture medium from each well was sampled and mixed with chemicals to achieve the final concentration, as described in the following. 

The toxic effect of oxalate was examined at concentrations of 0.1, 0.3, and 0.5 mM (prepared in 0.01 M phosphate-buffered saline, pH 7.4). The 0.5 mM concentration was chosen to examine the downstream signals of TRPV1, because this concentration is similar to that seen in the urine of 28-day hyperoxaluric rats. The concentrations of the following inhibitors were chosen based on their IC_50_ values: selective TRPV1 blockers capsazepine (Capz) 10 μM and SB-366791 (SB) 5 μM (Tocris, Minneapolis, MN, USA); specific ALOX12 inhibitor cinnamyl-3,4-dihydroxy-α-cyanocinnamate (CDC) 10 μM; membrane-permeant Ca^2+^ chelator 1,2-bis-(2-aminophenoxy)ethane-N,N,N′,N′-tetraacetic acid tetra(acetoxymethyl) ester (BAPTA) 20 μM; PKCα/β inhibitor Gö 6976 (Go) 5 μM; and selective Nox1/4 inhibitor GKT137831 (GKT) 10 μM. All chemicals, unless stated otherwise, were from Sigma-Aldrich (St Louis, MO, USA) and treated alone or in combination for 4, 8, or 24 h.

### 4.2. Evaluation of Cytotoxicity

A cytotoxicity kit (Roche Applied Science, Mannheim, Germany) was used to measure the amount of lactate dehydrogenase (LDH) release into culture medium, which indicates the degree of cell injury after treatment. The absorbance of the end reaction mixture was measured at 492 nm by a standard ELISA plate reader. The protein activity of LDH was quantitated against an LDH standard (Sigma-Aldrich), as previously described [10,26].

### 4.3. Ca^2+^ Image 

To study the effect of oxalate on changes in intracellular Ca^2+^ ([Ca^2+^]i) levels, cells were treated with 0.5 mM of oxalate in combination with TRPV1 blocker SB or Ca^2+^ chelator BAPTA for 24 h, as described previously [54]. After being washed with 0.01 M PBS (pH 7.4) 3 times, treated cells were incubated for 30 min with 4 μM of cell-permeant fluorescent dye Fluo-3 acetoxymethyl ester (Molecular Probes, Eugene, OR, USA) with 0.04% dimethyl sulfoxide (DMSO) and 0.02% pluronic acid at 37 °C. Cells were then imaged by an inverted microscope (Leica Microsystems GmbH, Wetzlar, Germany) equipped with a fluorescence image analytic system (Diagnostic Instruments, Sterling Heights, MI, USA) with the filter under excitation of 488 nm and emission of 522 nm to detect fluorescence intensity, as described previously [55].

### 4.4. Preparation of Culture Medium for ROS Determination

A series of culture medium samples (200 μL) were collected after treatment. The culture medium was immediately kept in the dark and on ice until chemiluminescence (CL) measurement, usually finished within 2 h, as described previously [54,56]. Before CL measurement, 0.1 mL of phosphate-buffered saline (pH 7.4) was added to 0.1 mL of culture medium. CL was measured in a completely dark chamber of the Chemiluminescence Analyzing System (CLD-110; Tohoku Electronic Industrial Co., Sendai, Japan). After 50 s background level determination, 1.0 mL of 0.1 mM lucigenin (Sigma-Aldrich) for detection of superoxide or 0.1 mM luminol (Sigma-Aldrich) for detection of H_2_O_2_ prepared in phosphate-buffered saline was injected into the sample. CL was monitored continuously for an additional 300 s. We previously showed that the CL recordings of lucigenin and luminol are derived from superoxide and H_2_O_2_, respectively [54,56]. For superoxide, the total amount of CL was calculated by integrating the area under the curve and subtracting it from the background level, expressed as CL counts per second. The amount of H_2_O_2_ in each sample was calculated using a standard curve prepared with concentrations of H_2_O_2_ ranging from 0.01 to 10 μM. These assays were performed in duplicate for each sample.

### 4.5. Experimental Animals

Male Wistar rats (~200 g) purchased from BioLASCO (the authorized distributor for Charles River Laboratories in Taiwan) were used in this study. Animal care and experimental protocols followed the Guide for the Care and Use of Laboratory Animals (National Academy Press, Washington, DC, USA, 2011).

### 4.6. Induction of Hyperoxaluria and CaOx Crystal Formation

Rats were housed at constant temperature with a 9 h light/15 h dark cycle (light from 08:00 to 17:00). The rats were divided into the following groups, which underwent treatment for 3–28 days. The control group received drinking water and standard chow. The hyperoxaluric rats were given standard chow mixed with 5% hydroxyl-L-proline (wt/wt HP/chow). The time point of 3 days means there is a hyperoxaluric state for the experimental animals but without renal deposition of CaOx crystals, and 7 to 28 days represents hyperoxaluria and CaOx deposition at various levels of severity in the rats [43,44]. During the experimental period, all rats had free access to food and water. The animals were placed in metabolic cages 2 days for acclimatization before collection of 24 h urine samples.

### 4.7. Preparation of Mini-Osmotic Pump for Chronic Drug Infusion

The mini-osmotic pumps (model 2004; Alzet, Cupertino, CA, USA) were incubated in saline at 37 °C for 4 h of activation before being implanted subcutaneously into the rats, according to the manufacturer’s instruction and our previous method, after rats were anesthetized by sodium pentobarbital (60 mg/kg i.p.) [57]. The pumps were carefully filled with selective TRPV1 inhibitor capsazepine (Capz) 80 mM, prepared in 50% dimethyl sulfoxide (DMSO) solution, or SB336791 (SB) 50 mM, prepared in 50% DMSO solution, by using a Hamilton syringe so as to avoid gas residue in the pump. The total amount of chemicals was released in 28 days at an optimal rate. All procedures were implemented under sterile conditions. The pumps were filled with the same volume of vehicle solution for the control group.

### 4.8. Metabolic Cage Study

After hyperoxaluric induction or drug treatment, rats were housed in individual metabolic cages with free access to water and food for monitoring of changes in body weight and food and water intake, and collection of 24 h urine, as described previously [26,43,44,54]. Urine was collected in tubes containing penicillin G (2000 IU) and streptomycin (2000 IU) to prevent microbial overgrowth. At the end of collection, rats were anesthetized by sodium pentobarbital (60 mg/kg) and perfused by cold 0.01M PBS (pH 7.4) via the transcardiac method [44], and the kidneys were removed, weighed, and stored at −80 °C or postfixed in paraformaldehyde for further analysis.

### 4.9. Urinalysis and Crystalluria

The urine volume was determined gravimetrically and expressed as mL/day, as previously described [26]. After determination, urine samples were centrifuged at 620 g for 10 min. The supernatant was separated from the sediment and collected for biochemical assays using commercially available ELISA kits. The sediment was dried over 48 h in an oven at 60 °C to measure dry weight. Urinary calcium level was measured using an electrolyte analyzer (Dri-Chem 3500i, Fuji, Tokyo, Japan), as described previously [44]. The magnesium concentration in the urine was determined using a commercial kit (BioAssay Systems, Hayward, CA, USA). Urinary oxalate and citrate levels were determined as described previously [44]. The degree of urine supersaturation was estimated according to the AP(CaOx) index using the formula (4076 × calcium^0.9^ × oxalate^0.96^)/[(citrate + 0.015)^0.60^ × magnesium^0.55^ × urine volume^0.99^)], as described previously [26]. Urinalysis was performed in duplicate.

### 4.10. CaOx Crystal Deposition

Tissue slices (5 μm thick) of rat kidney were prepared and stained with Pizzolato staining as previously described [26]. Briefly, kidney sections were deparaffinized and hydrated with distilled water. For staining, 30% hydrogen peroxide mixed with 5% silver nitrate (1 mL) was poured onto a slide with tissue sections (pH 6.0). The slide was exposed to light from a 60 W incandescent lamp at a distance of 15 cm for 20 min. Tissue sections were rinsed thoroughly with fresh mixture solution to remove gas bubbles during development. The slides were then washed thoroughly with distilled water, counterstained with nuclear fast red solution, and dehydrated in the usual manner. The extent of CaOx crystal deposition was graded semiquantitatively as follows: no crystal deposit (grade 0) to massive crystal deposits (grade 3) by a pathologist who was blind to the experiment.

### 4.11. Measurement of Cytokine Release and Tubular Damage

A commercial kit (R&D Systems, Inc., Minneapolis, MN, USA) was applied to measure the level of lL-1β in the culture medium and renal cortical tissues. The level of urinary kidney injury molecule-1 (KIM-1, a marker of tubular damage) was determined using a commercial ELISA kit (MD Biosciences, Oakdale, MN, USA) according to the protocol provided by the supplier.

### 4.12. Western Blot Analysis

Cells and renal cortical tissues were collected and prepared as cytosolic fraction, membrane fraction, or total protein using a commercial kit (BioVision, Milpitas, CA, USA) as described previously [26]. Protein samples were quantitated using a commercial assay kit (Bio-Rad, Hercules, CA, USA). Samples were then separated on SDS polyacrylamide gels under denaturing conditions and electrophoretically transferred to polyvinylidenedifluoride membrane (Amersham-Pharmacia Biotech, Little Chalfont, UK). After blocking in 5% skimmed milk, the membranes were incubated overnight at 4 °C with antibodies against TRPV1, ALOX12, PKCα, Nox2, Nox4, NLRP3, IL-1β, caspase-1, actin, or Na^+^, K^+^-ATPase (Santa Cruz Biotechnology, Santa Cruz, CA, USA). After washing, the membranes were incubated for 1 h at room temperature with a corresponding horseradish peroxidase-conjugated IgG (Jackson ImmunoResearch, West Grove, PA, USA). After washing the membranes, the bound antibody complex was detected using a commercial enhanced chemiluminescence kit (Thermo Scientific, Rockford, IL, USA). The density of the bands of appropriate molecular masses was determined semi-quantitatively by densitometry using an image analytic system (Diagnostic Instruments, Sterling Heights, MI, USA). The level of each protein was expressed relative to the amount of actin or Na^+^, K^+^-ATPase.

### 4.13. Localization of ALOX12 or TRPV1 Expression

Indirect immunofluorescent staining was used to examine the cellular distribution of ALOX12 or TRPV1 in tubular cells or renal tissues, as described previously [25]. Briefly, postfixed kidneys were stored in 10% sucrose solution containing 4% paraformaldehyde at 4 °C, embedded in O.C.T. compound (Tissue-Tek, Sakura Finetek, Torrence, CA, USA), and frozen at −20 °C, then cut into 5 μm sections on a cryostat (Microm, Heidelberg, Germany), which were thaw-mounted on coated slides. PTCs were cultured on poly-D-lysine-precoated glass dishes and postfixed in 4% paraformaldehyde solution at 4 °C for 2 h. After being rehydrated and washed with 0.01 M PBS (pH 7.4), tissue sections and culture dishes were processed for indirect immunofluorescence using a tyramide signal amplification kit (PerkinElmer, Waltham, MA, USA). After being blocked with 5% skimmed milk prepared in PBS for 1 h, sections were incubated overnight at 4 °C with an antibody against ALOX12 or TRPV1 and then for 1 h at room temperature (RT) with the corresponding cyanine 3- (for tubular cells) or fluorescein- (for tissue slides) conjugated secondary antibody, and examined on an inverted fluorescent microscope, as described above. Nuclei were counterstained using DAPI. The specificity of each antibody was tested by preincubation with the specific blocking peptide provided by Santa Cruz Biotechnology (150 μg/mL) before performing the test.

### 4.14. Quantitative Real-Time PCR for Nox Expression in Tubular Cells

RNA was extracted using a commercial kit (RareRNA, Bio-East Technology, Taipei, Taiwan), as previously described [56], and cDNA was synthesized at 42 °C for 45 min (reaction mixture: 2 μg of RNA, 5 μg of poly(dT)15 oligonucleotide primer (Life Technologies, Waltham, MA, USA)) and exposed to 200 units of reverse transcriptase (Moloney murine leukemia virus; Promega, Madison, WI, USA). Quantitative RT-PCR was performed in an ABI StepOne Plus system (Applied Biosystems, Foster City, CA, USA). PCR was performed using 100 ng of cDNA and 30 μmol of primer (total reaction volume, 20 μL) with the SYBR Green PCR master mix kit, according to the manufacturer’s instructions (Applied Biosystems). The following primers were used for PCR as Nox2: 5′-CCA GTG AAG ATG TGT TCA GCT -3′ (forward) and 5′-GCA CAG CCA GTA GAA GTA GAT-3′ (reverse); Nox4: 5′-GGA TCA CAG AAG GTC CCT AGC-3′ (forward) and 5′-AGA AGT TCA GGG CGT TCA CC-3′ (reverse); glyceraldehyde 3-phosphate dehydrogenase (GAPDH): 5′-TTA GCA CCC CTG GCC AAG G-3′ (forward) and 5′-CTT ACT CCT TGG AGG CCA TG-3′ (reverse) [42,54]. The cycling conditions were as follows: 95 °C for 20 s, followed by 40 cycles at 95 °C for 1 s and 60 °C for 20 s. A melting curve analysis was performed at the end of each PCR experiment. All reactions were run in duplicate. The ΔCt (threshold cycle) was calculated by subtracting the raw Ct values for GAPDH from the raw Ct values for the target genes, thereby providing information about relative changes in gene expression. Changes in TRPV1 and Nox expression were calculated as 2^−ΔCt^ and expressed as fold change relative to the control group.

### 4.15. Measurement of Caspase-1 Activity

Caspase-1 activity in cell lysates and renal cortical tissues was determined using a commercial fluorometric assay kit (ab39412) obtained from Abcam (Bristol, UK) and expressed as units/mg of protein.

### 4.16. Statistics

Numerical data were presented as mean ± SEM. Differences between groups were analyzed using an unpaired *t*-test or one-way ANOVA, with a post-test using Duncan’s multiple-range test. Differences were regarded as significant when *p* < 0.05.

## Figures and Tables

**Figure 1 ijms-22-06204-f001:**
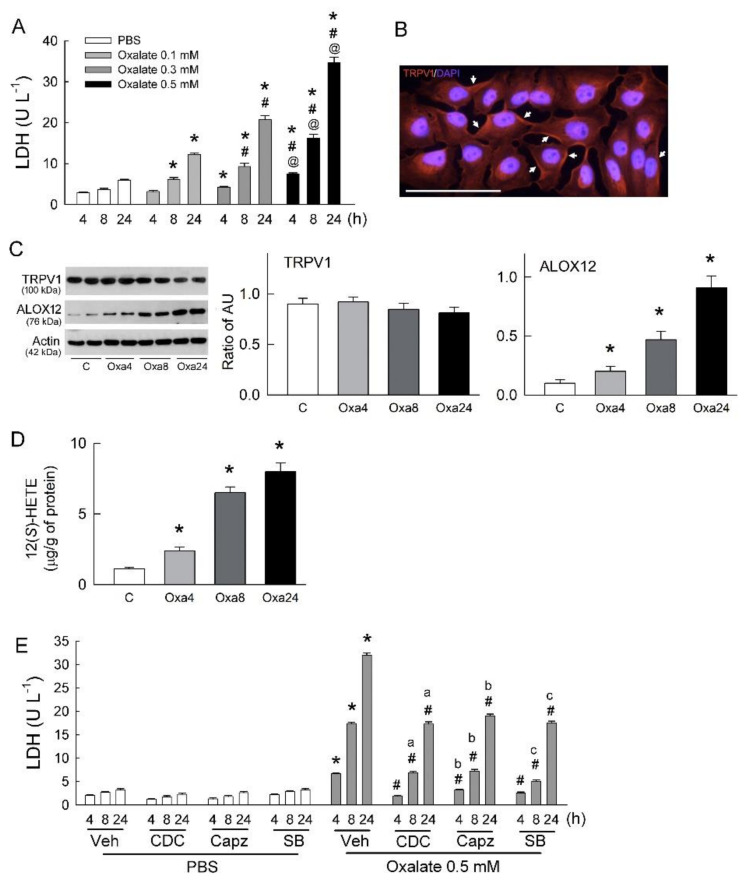
Transient receptor potential vanilloid 1 (TRPV1) inhibition abrogates oxalate-mediated tubular cell injury and increases in cytosolic Ca^2+^. (**A**) Release of lactate dehydrogenase (LDH) into culture medium, determined as a marker of cell injury after LLC-PK_1_ cells treated with various doses of oxalate for 4, 8, and 24 h (n = 6 for each dose and time point). * *p* < 0.05 vs. PBS-treated group (control) at the same time-point; ^#^
*p* < 0.05 vs. oxalate 0.1 mM group at the same time-point; ^@^
*p* < 0.05 vs. oxalate 0.3 mM group at the same time-point. One-way ANOVA was applied for statistical comparison. (**B**) Representative image showing presence of TRPV1 (in red, indicated by arrows) in cells. Cell nuclei were counterstained by 4’,6-diamidino-2-phenylindole (DAPI, blue). White bar indicates scale of 300 μm. (**C**) Representative blots show TRPV1 and ALOX12 expression (n = 2 per group) in LLC-PK_1_ cells using 20 μg of total protein in each lane. Right bar graph shows densitometrically quantified results by ratio of TRPV1 or ALOX12 to actin (n = 6 per group). AU, arbitrary unit of band density; C, control cells treated with PBS for 24 h; Oxa4, Oxa8, and Oxa24 are indicated cells treated with 0.5 mM oxalate for 4, 8, and 24 h, respectively. * *p* < 0.05 vs. control (C) group analyzed by unpaired *t*-test. (**D**) Cellular content of 12(*S*)-HETE in groups. * *p* < 0.05 versus C group analyzed by unpaired *t*-test. (**E**) Release of LDH after cells treated with PBS or 0.5 mM oxalate or in combination with blockers of ALOX12 (CDC) or TRPV1 (Capz and SB) for 4, 8, and 24 h (n = 6 for each group and time point). Veh, 0.1% DMSO as vehicle solution. Note that releases of LDH in the CDC, Capz, and SB plus oxalate groups for 4, 8, and 24 h of cotreatment were all significantly lowered when compared to those in the oxalate-treated group. * *p* < 0.05, oxalate- vs. PBS-treated group at the same time-point; ^a^
*p* < 0.05, CDC + oxalate group vs. CDC + PBS group at the same time-point; ^b^
*p* < 0.05, Capz + oxalate group vs. Capz + PBS group at the same time-point; ^c^
*p* < 0.05, SB + oxalate group vs. SB + PBS group at the same time-point; ^#^
*p* < 0.05 vs. 0.5 mM oxalate-treated group at the same time-point. One-way ANOVA was applied for statistical comparison.

**Figure 2 ijms-22-06204-f002:**
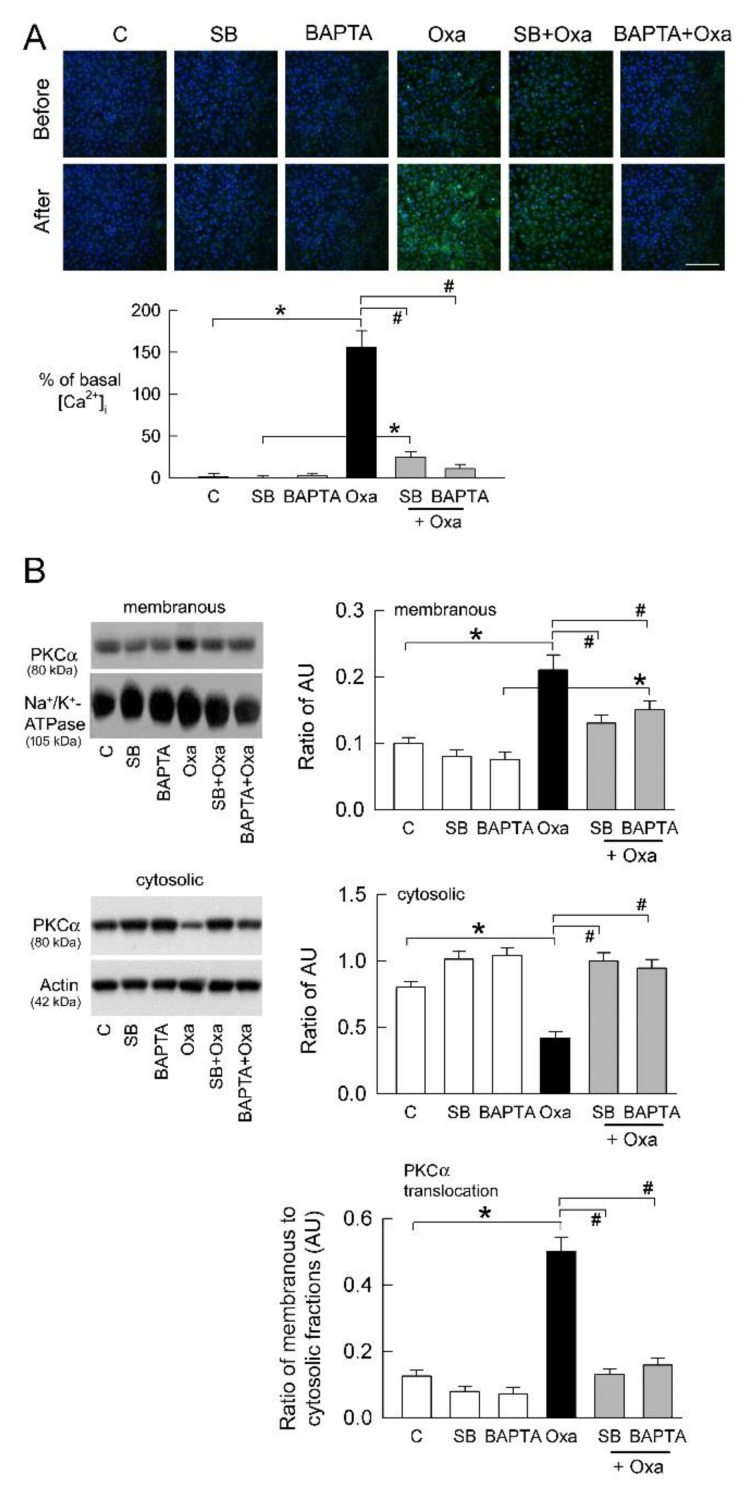
TRPV1 inhibition and Ca^2+^ chelation attenuates oxalate-mediated PKCα activation in tubular cells. (**A**) Example of merged pictures of Fluo-3-loaded LLC-PK_1_ cells with various treatments at 200× magnification (green). Nuclei are counterstained with DAPI (blue). White scale bar is 800 μm. Lower bar graph shows percent changes in [Ca^2+^]i of control (C) group (n = 6 per group). Note that a significant increase in Fluo-3 fluorescence following 24 h of 0.5 mM oxalate (Oxa) treatment is attenuated by TRPV1 blocker SB and Ca^2+^ chelator BPATA. C, control. (**B**) Representative blots are shown for PKCα in cell membrane (right upper blots) and cytosolic (right lower blots) fraction. Equal amounts of 40 μg protein per lane from each sample were loaded, as confirmed by similar expression of Na^+^, K^+^-ATPase, and actin in membranous and cytosolic fractions, respectively. Relative levels of PKCα were assessed by densitometry as ratio of PKC to Na^+^, K^+^-ATPase (left upper bar graph) or actin (left middle bar graph). Note that oxalate induces a significant increase in membranous distribution and a reduced cytosolic fraction of PKCα. These were attenuated by TRPV1 blocker SB and Ca^2+^ chelator BPATA. * *p* < 0.05, Oxa vs. control (C) group; ^#^
*p* < 0.05 vs. Oxa group without blocker treatment. One-way ANOVA was applied for statistical comparison in all graphs.

**Figure 3 ijms-22-06204-f003:**
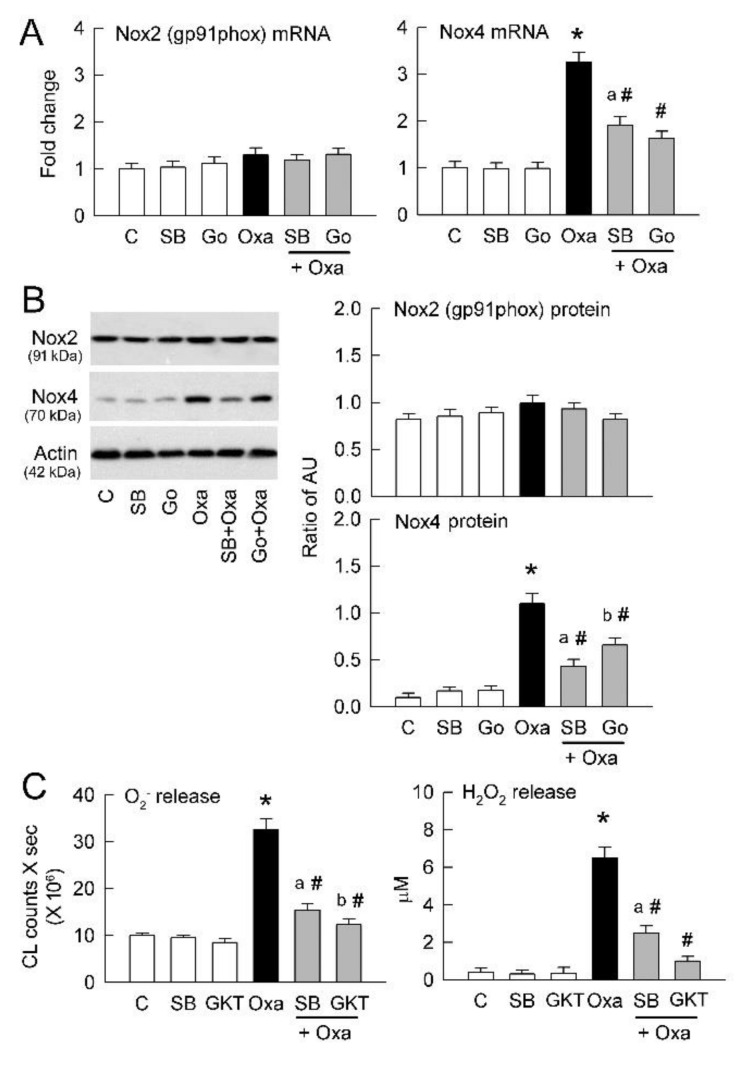
Blockade of TRPV1 and PKC on oxalate-mediated Nox expression and reactive oxygen species (ROS) generation in tubular cells. (**A**) mRNA levels of Nox2 (as gp91phox) and Nox4 were examined by real-time quantitative PCR and expressed as fold change of control (C) cells (n = 6 per group). (**B**) Representative blots are shown for Nox2 and Nox4 expression using 10 μg of total protein in each group. Right bar graph shows densitometrically quantified results by ratio of Nox2 or Nox4 to actin (n = 6 per group). AU, abritary unit of band density. Note that TRPV1 blocker SB-366791 (SB) and PKC blocker Go-6976 (Go) attenuate an oxalate-induced increase in Nox4 expression. (**C**) Right bar graph shows total amount of lucigenin-dependent chemiluminescence (CL) for superoxide generation as area under the curve (n = 6 in each group). Left bar graph shows the amount of H_2_O_2_ in culture medium in groups (n = 6 in each group). Note that inhibition of TRPV1 by SB or Nox by GKT-137831 (GKT) significantly attenuates oxalate-mediated superoxide and H_2_O_2_ formation. * *p* < 0.05 vs. corresponding control group without oxalate treatment; ^#^
*p* < 0.05 vs. oxalate group. * *p* < 0.05, oxalate (Oxa) vs. control (C) group; ^a^
*p* < 0.05, SB + Oxa group vs. SB group; ^b^
*p* < 0.05, GKT + Oxa group vs. GKT group; ^#^
*p* < 0.05 vs. Oxa group. One-way ANOVA was applied for statistical comparison in all graphs.

**Figure 4 ijms-22-06204-f004:**
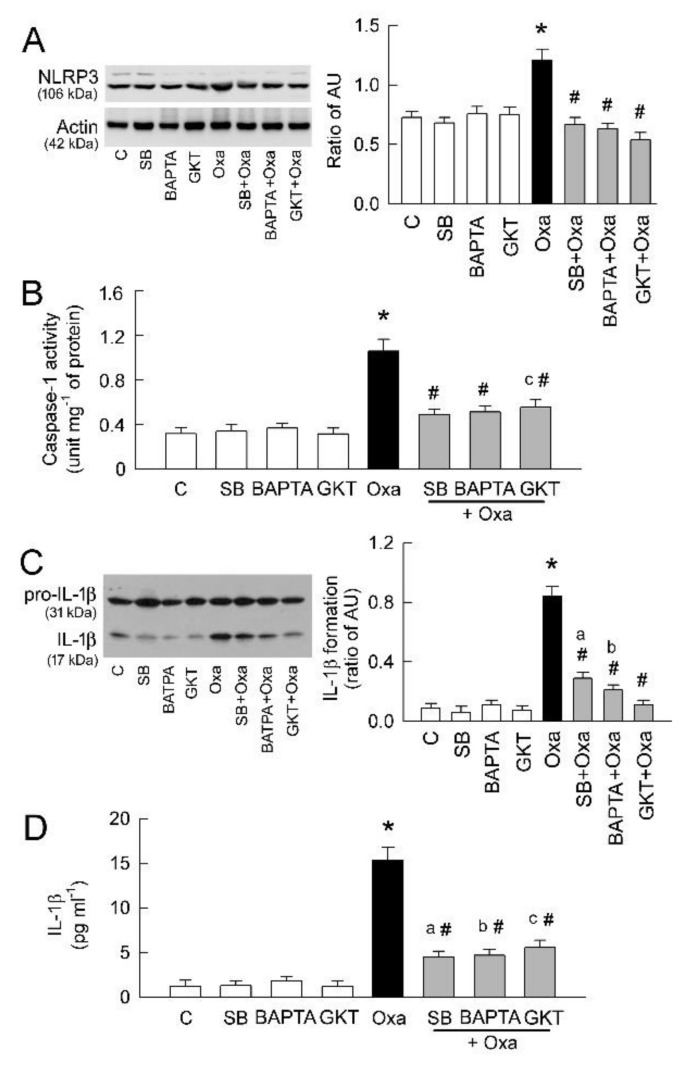
Inhibition of TRPV1 atteuates oxalate-mediated inflammatory response in tubular cells. (**A**) Representative blots are shown for NLRP3 expression using 10 μg of total protein in each group. Right bar graph shows densitometrically quantified results by ratio of NLRP3 to actin (n = 6 per group). AU, abritary unit of band density. Note that TRPV 1 blocker SB, Ca^2+^ chelator BPATA, and Nox inhibitor GKT totally abrogate NLRP3 upregulation caused by oxalate (Oxa). (**B**) Bar graph showing caspase-1 activity measured in cell lysates after 24 h of various treatments. Besides TRPV1 inhibition by SB, co-treatment of Ca^2+^ chelator BPATA and Nox inhibitor GKT with oxalate also totally abrogates oxalate-induced increases in caspase-1 activity. (**C**) Representative blots are shown for pro-IL-1β (at 31 kDa) and mature IL-1β (at 17 kDa) expression using 20 μg of total protein in each group. Right bar graph shows densitometrically quantified results by ratio of IL-1β to pro-IL-1β to represent IL-1β formation in cells (n = 6 per group). AU, abritary unit of band density. Note that TRPV1 blocker SB, Ca^2+^ chelator BPATA, and Nox inhibitor GKT attenuate oxalate-mediated IL-1β formation. (**D**) Quantitative results showing IL-1β release into culture medium after 24 h of various treatments (n = 6 per group). * *p* < 0.05, oxalate (Oxa) vs. control (C) group; ^a^
*p* < 0.05, SB + Oxa group vs. SB group; ^b^
*p* < 0.05, BPATA + Oxa group vs. BPATA group; ^c^
*p* < 0.05, GKT + Oxa group vs. GKT group; ^#^
*p* < 0.05 vs. Oxa group. One-way ANOVA was applied for statistical comparison in all graphs.

**Figure 5 ijms-22-06204-f005:**
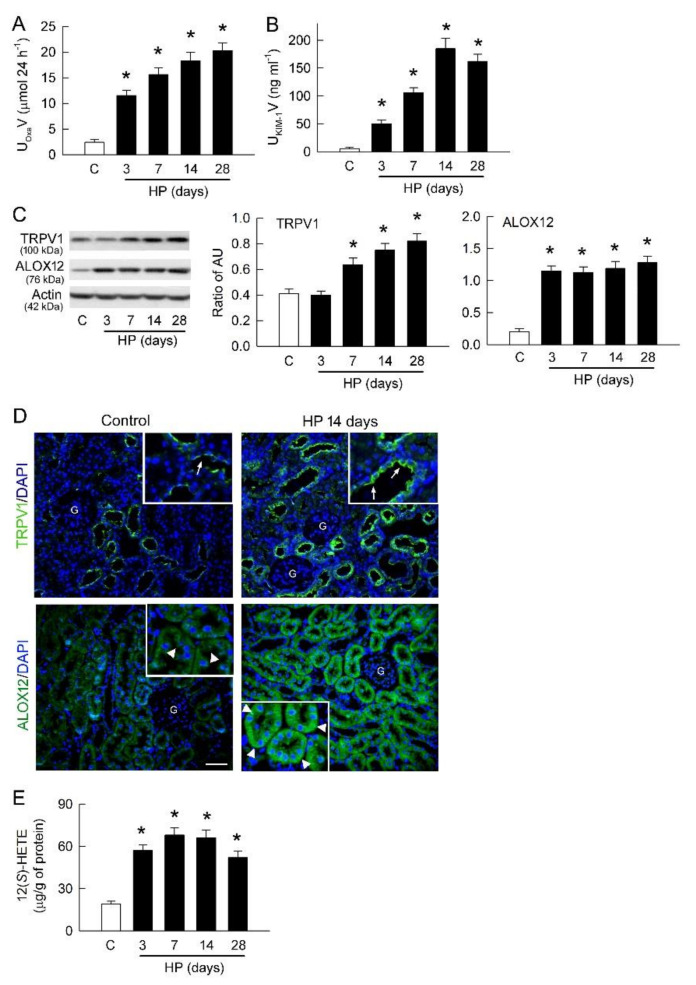
Effect of hyperoxaluria on renal TRPV1, ALOX12, and 12(*S*)-HETE expression. (**A**) Urine samples were collected daily from metabolic cage study to measure excreted oxalate after hyperoxaluric (HP) induction. Urine in control (C) rats was collected only on day 28. Note that hyperoxaluria increases in a time-dependent manner in HP rats. (**B**) Urinary KIM-1 level, as a marker for tubular damage, demonstrates an increase in HP rats at all induction times. (**C**) Representative blots are shown for TRPV1 and ALOX12 expression using 10 μg of total protein in each lane. Right bar graph shows densitometrically quantified results by ratio of TRPV1 or ALOX12 to actin. AU, abritary unit of band density. (**D**) Representative pictures show tissue localization of TRPV1 (upper two pictures) and ALOX12 (lower two pictures) in renal cortex of control and 14-day HP kidneys at 200× magnification. Cell nuclei are counterstained with DAPI (blue). G, glomerulus. The local tissue of the renal cortex was magnified at 400× (insets) and demonstrated that TRPV1 and ALOX12 (green) are present at apical membrane (indicated by arrows) and cytosol (indicated by arrow heads) of renal tubules, respectively, in control and HP (14 days) kidneys. These expressions are increased in HP kidney. (**E**) Tissue content of 12(*S*)-HETE in renal cortex; n = 6 for each group and time point for all bar graphs. * *p* < 0.05, HP vs. control by unpaired *t*-test.

**Figure 6 ijms-22-06204-f006:**
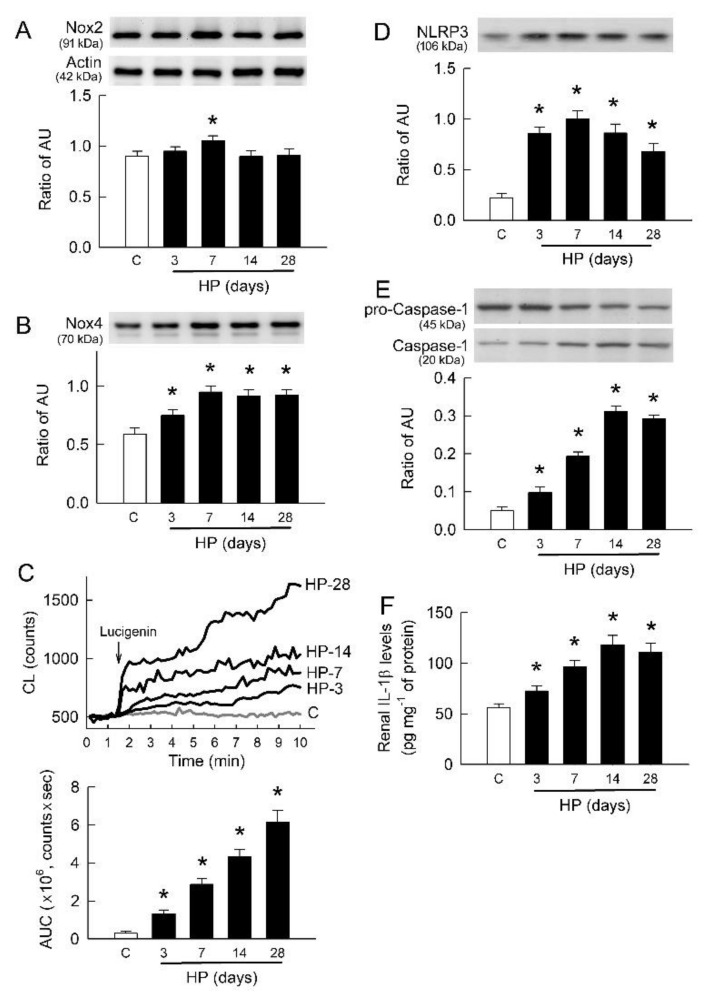
Hyperoxaluria increases Nox4, oxidative stress, and inflammatory mediator expression. (**A**,**B**) Representative blots show protein expression of Nox2 (**A**) and Nox4 (**B**) (10 μg of total protein per lane) at appropriate molecular weights. Lower bar graphs (n = 6 in each group) show ratio of abritary unit in band density of Nox2 or Nox4 to actin at 6A. (**C**) Typical tracings show changes in lucigenin-dependent chemiluminescence (CL) counts for superoxide formation in kidney surface of control (C) and 3- to 28-day HP rats. Lower bar graph shows total amount of CL as area under the curve (AUC) for n = 6 in each group and time point. (**D**) Representative blots show protein expression of NLRP3 (10 μg of total protein per lane). Lower bar graphs (n = 6 per group and time point) show ratio of band density of NLRP3 to actin at 6A. (**E**) Representative blots show pro-caspase-1 (precursor at 45 kDa) and caspase-1 (at 17 kDa) expression using 10 μg of total protein in each group. Lower bar graph shows densitometrically quantified results by ratio of caspase-1 to pro-caspase-1 to represent caspase-1 activity (n = 6 per group). (**F**) Quantitative results showing increased tissue content of IL-1β in renal cortex of HP rats (n = 6 per group). * *p* < 0.05, HP vs. control group analyzed by unpaired *t*-test.

**Figure 7 ijms-22-06204-f007:**
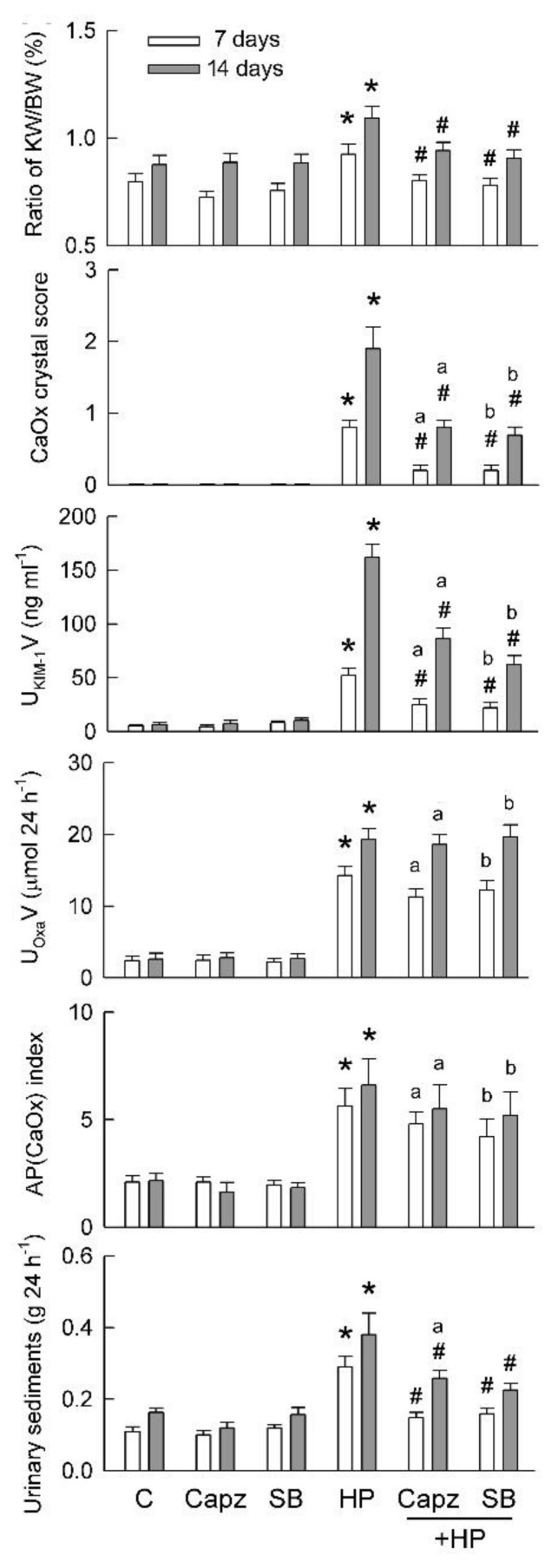
Chronic TRPV1 inhibition abolishes effect of hyperoxaluria on kidney injury and CaOx crystal formation. Basic data were obtained from metabolic cage study and collected over 24 h on day 7 and 14. Both sides of kidneys were collected to calculate percent changes in ratio of kidney to body weight (KW/BW). Deposition of CaOx crystals was examined and scored after staining. Urine was analyzed for KIM-1 (U_KIM-1_V) and oxalate (U_Oxa_V) excretion. Degree of urinary supersaturation was estimated according to AP(CaOx) index. Precipitated particles in urine were dried to determine weight of urine sediments. N = 6 in each group and time point. Capz and SB are TRPV1 blockers. * *p* < 0.05, HP vs. control (C) group on the same induction day. ^a^
*p* < 0.05, Capz + HP group vs. HP group on the same induction day; ^b^
*p* < 0.05, SB + HP group vs. HP group at the same induction day; ^#^
*p* < 0.05, blocker-treated vs. untreated HP group on the same induction day. One-way ANOVA was applied for statistical comparison in all graphs.

**Figure 8 ijms-22-06204-f008:**
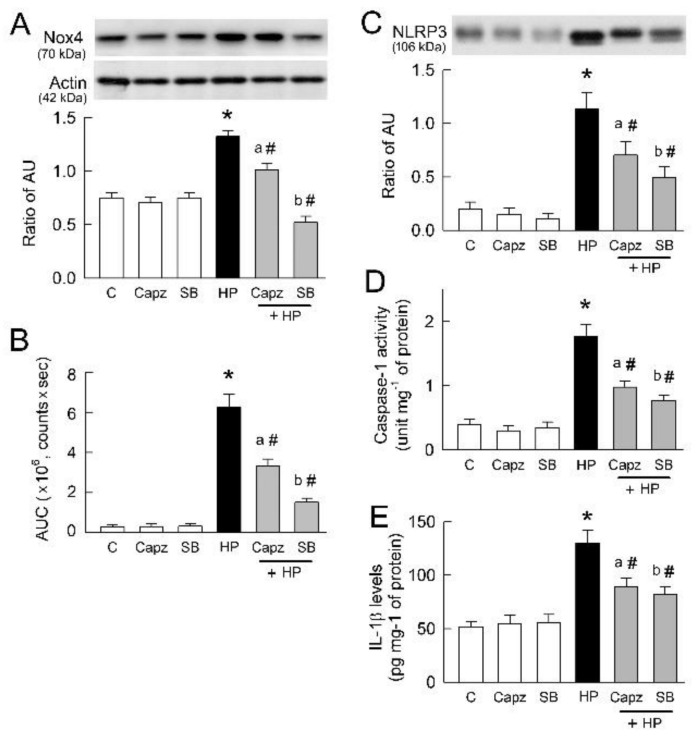
Chronic TRPV1 inhibition attenuates hyperoxaluria-induced oxidative stress and inflammation on day 14. (**A**) Representative blots show expression of Nox4 using 10 μg of total protein in each group. Lower bar graph shows densitometrically quantified results by ratio of Nox4 to actin (n = 6 per group). (**B**) Bar graph shows total chemiluminescence (CL) counts for superoxide formation in kidney surface as area under the curve (AUC) for n = 6 in each group. Note that TRPV1 inhibition significantly attenuates oxidative stress in HP kidney. (**C**) Representative blots show expression of NLRP3 using 10 μg of total protein in each group. Lower bar graph shows densitometrically quantified results by ratio of NLRP3 to actin at 8A (n = 6 per group). (**D**) Bar graph showing caspase-1 activity measured in tissues of renal cortex in group. (**E**) Quantitative results showing increased tissue content of IL-1β in group (n = 6 per group). Note that TRPV1 inhibition significantly attenuates renal inflammation caused by hyperoxaluria. * *p* < 0.05, HP vs. control (C) group. ^a^
*p* < 0.05, Capz + HP group vs. HP group; ^b^
*p* < 0.05, SB + HP group vs. HP group; ^#^
*p* < 0.05, blocker-treated vs. untreated HP group. One-way ANOVA was applied for statistical comparison in all graphs.

**Figure 9 ijms-22-06204-f009:**
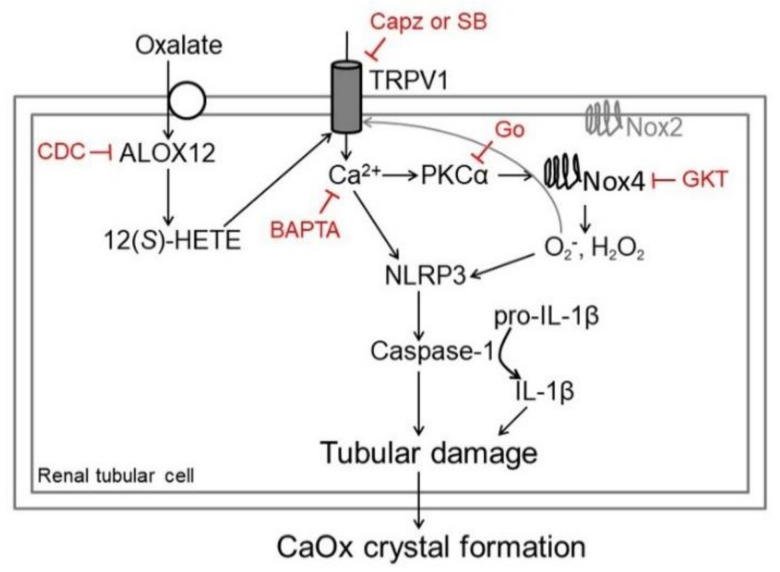
Schematic diagram showing how TRPV1 hyperfunction contributes to tubular damage in hyperoxaluria. After uptake by tubular cells, oxalate overstimulates TRPV1 via an enhanced function of arachidonate 12-lipoxygenase (ALOX12) on synthesis of endovanilloid 12(*S*)-hydroxyeicosatetraenoic acid [12(*S*)-HETE]. Blockade of ALOX12 by cinnamyl-3,4-dihydroxy-α-cyanocinnamate (CDC) decreases 12(*S*)-HETE. TRPV1 hyperfunction induces intracellular Ca^2+^ elevation, increasing PKCα activity, as confirmed by Ca^2+^ chelating effect of BAPTA. PKCα subsequently triggers NADPH oxidase (Nox) 4 but not Nox2 activation, which is attenuated by PKC inhibition (Gö 6976, Go). Superoxide (O_2_^−^) and hydrogen peroxide (H_2_O_2_) are liberated and result in oxidative stress, attenuated by Nox inhibition (GKT137831, GKT). ROS generated by Nox4 may also activate TRPV1. Tubular damage is further evidenced by increased inflammasome NLR family pyrin domain-containing 3 (NLRP3) expression, caspase-1 activation, and maturation of IL-1β for secretion. Besides TRPV1 inhibition, blockade of more downstream effectors such as Nox4-mediated oxidative stress and Ca^2+^ chelation attenuates oxalate- or hyperoxaluria-induced tubular cell inflammation.

## Data Availability

Data is contained within the article.

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
