# Peer review of "TRPV1 Hyperfunction Contributes to Renal Inflammation in Oxalate Nephropathy"

_ijms, 2021, doi:10.3390/ijms22126204_

Round 1

Reviewer 1 Report

The manuscript entitled, “TRPV1 Hyperfunction Contributes to Renal Inflammation in Oxalate Nephropathy” is an extremely comprehensive and very well written and organized manuscript attempting to show how oxalate increases TRPV1 function in the proximal tubule, which increases inflammatory marker expression, ROS production, and ultimately tubular dysfunction. However, a few major clarifications and adjustments should be addressed before consideration for publication:

  1. Overall, making the leap that TRPV1 inhibition attenuates calcium oxalate (CaOx) formation in this oxalate-induced nephropathy model is not conclusive. While it is significantly evident that TRPV1 plays a role in oxalate-induced inflammation and proximal tubule cell dysfunction, the only CaOx crystal measurement data was collected in the 5th graph of Figure 7, which actually shows that the TRPV1 inhibitors have no effect on CaOx crystal formation because you see a significant increase at both time points in all oxalate-treated groups regardless of TRPV1 inhibitor treatment. Was this CaOx index verified and determined by a blinded pathologist?

The conclusions of the manuscript should be focused on the oxalate only-induced inflammation in the tubules being mediated by TRPV1 function – to tie this to crystallization is not appropriate given the data.

Additionally, one could hypothesize that the function of this calcium channel may actually be protective against crystallization since it’s an apically-localized calcium importer, which in the presence of oxalate, would be upregulated to remove calcium from the lumen to prevent its complex with oxalate. Therefore, delineating these results and conclusions from kidney stones and focusing on direct oxalate-induced inflammation and damage is warranted.

  1. There is a concern that body weight measurements alone of the rats treated with HP over the course of the 28 days of treatment to monitor direct HP-induced toxicity were not included. Given the KIM-1 results, it seems these animals were experiencing significant renal impairment, which could significantly affect Figure 7 results.
  2. In Figure 1E, comparisons of the TRPV1 inhibitors at the 3 different time points is not clear in terms of which time is being used as the comparator in the vehicle, oxalate-alone group, i.e. do the symbols at 24 h in the CDC, Capz, and SB groups indicate a significant decrease in LDH release when compared to the 24 h veh oxalate-alone group?

Additionally, many of the graphs have symbols indicating significant differences from their control, inhibitor-only groups with no oxalate indicating that even in the presence of the inhibitors, oxalate it still capable of significantly inducing a toxic/inflammatory effect. For example, Figure 4D.

Author Response

Response to Reviewer 1’s comments

Overall, making the leap that TRPV1 inhibition attenuates calcium oxalate (CaOx) formation in this oxalate-induced nephropathy model is not conclusive. While it is significantly evident that TRPV1 plays a role in oxalate-induced inflammation and proximal tubule cell dysfunction, the only CaOx crystal measurement data was collected in the 5th graph of Figure 7, which actually shows that the TRPV1 inhibitors have no effect on CaOx crystal formation because you see a significant increase at both time points in all oxalate-treated groups regardless of TRPV1 inhibitor treatment. Was this CaOx index verified and determined by a blinded pathologist?

Response: Thank you for your comments. The CaOx crystal scores were examined by a pathologist who was blind to the experiment. We amended this description in the section of Materials and Methods. Please see the section 4.10 and line 622.

As per your comments, TRPV1 inhibition did not affect high degree of supersaturation [as estimated by AP(CaOx) index] caused by hyperoxaluria in this study. In the same figure, the second bar graph however showed that TRPV1 inhibition significantly attenuated CaOx crystal deposition in the hyperoxaluric rats by lowering crystal scores. This result clearly indicated that TRPV1 inhibition have a beneficial effect on CaOx crystal formation in the rat model. During CaOx crystal formation, urinary supersautration seems not enough for crystallization. Formation of CaOx crystal begins from the attachment of a crystal formed in the cavity of renal tubules to the surface of renal tubular cells [Tsujihata, 2008]. Moreover, renal tubular cells are injured by hyperoxaluria and crystals easily become attached to them [Tsujihata, 2008]. Thus, the cell debris of injured tubular cells caused by hyperoxaluria favors CaOx crystal formation as a seed in crystal growth and aggregation [Tseng et al., 2020]. Here we showed TRPV1 inhibition attenuates oxalate- or hyperoxaluria-induced tubular cell damage, this may prevent CaOx crystal formation by inhibition of crystallization and a further crystal growth after aggregation.

References

  1. Tsujihata M. Mechanism of calcium oxalate renal stone formation and renal tubular cell injury. Int J Urol. 2008 Feb;15(2):115-20. doi: 10.1111/j.1442-2042.2007.01953.x.
  2. Tseng YS, Wu WB, Chen Y, Lo Yang F, Ma MC. Small intestine resection increases oxalate and citrate transporter expression and calcium oxalate crystal formation in rat hyperoxaluric kidneys. Clin Sci (Lond). 2020 Oct 16;134(19):2565-2580. doi: 10.1042/CS20200973.

The conclusions of the manuscript should be focused on the oxalate only-induced inflammation in the tubules being mediated by TRPV1 function – to tie this to crystallization is not appropriate given the data.

Response: Thank you for your comments. Most of our conclusion are focused on oxalate- or hyperoxaluria-mediated inflammation. As explained above, our results in Figure 7 showed that TRPV1 inhibition significantly attenuates CaOx crystal deposition in the hyperoxaluric rats. We therefore related oxalate- or hyperoxaluria-induced inflammation as observed in our in vitro and in vivo experiments to CaOx crystal formation. This was consistent with previous findings showed that renal tissue inflammation plays a pivotal role in CaOx crystal formation [Khan, 2014; Joshi et al., 2015].

References

  1. Khan SR. Reactive oxygen species, inflammation and calcium oxalate nephrolithiasis. Transl Androl Urol. 2014 Sep 1;3(3):256-276. doi: 10.3978/j.issn.2223-4683.2014.06.04.
  2. Joshi S, Wang W, Peck AB, Khan SR. Activation of the NLRP3 inflammasome in association with calcium oxalate crystal induced reactive oxygen species in kidneys. J Urol. 2015 May;193(5):1684-91. doi: 10.1016/j.juro.2014.11.093.

Additionally, one could hypothesize that the function of this calcium channel may actually be protective against crystallization since it’s an apically-localized calcium importer, which in the presence of oxalate, would be upregulated to remove calcium from the lumen to prevent its complex with oxalate. Therefore, delineating these results and conclusions from kidney stones and focusing on direct oxalate-induced inflammation and damage is warranted.

Response: Thank you for your valuable comments. Actually our original idea about TRPV1 was the same as your comments that enhanced its function in proximal tubules may attenuate hypercalciuria and thus benefits hyperoxaluric kidney to reduce CaOx crystal formation. We ever tested the effect of capsaicin (a TRPV1 agonist) on oxalate-treated cells as well as in the HP rats and showed that capsaicin exacerbates oxalate- or hyperoxaluria-mediated tubular damage in both in vitro and in vivo studies. These pilot studies turn our thinking that TRPV1 is not same as TRPV5 which presents in renal connecting tubule and distal tubule where TRPV5 is simply responsible for calcium reabsorption to attenuate hyperoxaluria. We therefore interested to know the exact role of TRPV1 in crystal formation, especially literatures have reported a pro-inflammatory role of TRPV1. Your valuable comments were included in the section of Discussion. Please see page 14, lines 391-395.

There is a concern that body weight measurements alone of the rats treated with HP over the course of the 28 days of treatment to monitor direct HP-induced toxicity were not included. Given the KIM-1 results, it seems these animals were experiencing significant renal impairment, which could significantly affect Figure 7 results.

Response: Thank you for your comments. In this study, the body weight of HP rats after 28 days of induction was 321±6.7 g, which slightly lowered but insignificantly than 336±4.2 g in the control rats. This result was consistent with previous findings [Huang et al., 2008; Huang & Ma, 2015]. The body weight of HP rats was even not changed too much after 42 days of hyperoxaluric induction. Concerning the urinary level of KIM-1, we believe that there is a renal insufficiency in HP rats. This however did not affect gain of body weight after 28 days of adaptation. Therefore, the increase in the ratio of kidney to body weight shown in the first bar graph of Figure 7 is simply due to the increased kidney weight caused by CaOx crystal deposition and tissue damage/regeneration.

References

  1. Huang HS, Ma MC, Chen J. Chronic L-arginine administration increases oxidative and nitrosative stress in rat hyperoxaluric kidneys and excessive crystal deposition. Am J Physiol Renal Physiol. 2008 Aug;295(2):F388-96. doi: 10.1152/ajprenal.00405.2007.
  2. Huang HS, Ma MC. High Sodium-Induced Oxidative Stress and Poor Anticrystallization Defense Aggravate Calcium Oxalate Crystal Formation in Rat Hyperoxaluric Kidneys. PLoS One. 2015 Aug 4;10(8):e0134764. doi: 10.1371/journal.pone.0134764.

In Figure 1E, comparisons of the TRPV1 inhibitors at the 3 different time points is not clear in terms of which time is being used as the comparator in the vehicle, oxalate-alone group, i.e. do the symbols at 24 h in the CDC, Capz, and SB groups indicate a significant decrease in LDH release when compared to the 24 h veh oxalate-alone group?

Response: Thank you for the comment. We have added bars on the top the bar graphs to show the statistic significance between groups as per another Reviewer’s comments. In Figure 1E, star signs indicate there is significant increases in LDH releases in the oxalate-treated groups when compared to those in the corresponding PBS-treated (control) groups. Pound signs indicate a significant increase in LDH release in the inhibitor plus oxalate groups as compared to that in the oxalate-treated group. These symbols were only marked when the statistic comparisons are performed at the same time-point. The releases of LDH in the CDC, Capz, and SB plus oxalate groups for 4, 8, and 24 h of cotreatment were all significantly lowered when compared to those in the oxalate-treated group. We have included this description in the legend of Figure 1. Please see page 4, lines 119-120.

Additionally, many of the graphs have symbols indicating significant differences from their control, inhibitor-only groups with no oxalate indicating that even in the presence of the inhibitors, oxalate it still capable of significantly inducing a toxic/inflammatory effect. For example, Figure 4D.

Response: Thank you for your comments. In Figure 4, some effects of blockers (SB, BAPTA, or GKT) were partially antagonized the effect of oxalate on inflammatory activation. This can be explained there is a pathway other than we studied here is present to affect inflammatory response. For example, in Figure 4A all blockers were totally abrogated NLRP3 upregulation caused by oxalate. As per your comment, in Figure 4D these blockers however were partially attenuate oxalate-mediated IL-1β secretion. Previous study has reported other inflammasomes such as NLRC4 and NLRP1, and DNA sensing receptors AIM2 and RIG-I can also induce IL-1β secretion [Lopez-Castejon & Brough, 2011]. Whether these molecules involved in oxalate-mediated IL-1β release require a further study.

Reference

  1. Lopez-Castejon G, Brough D. Understanding the mechanism of IL-1β secretion. Cytokine Growth Factor Rev. 2011 Aug;22(4):189-95. doi: 10.1016/j.cytogfr.2011.10.001.

Reviewer 2 Report

In this manuscript authors tested the possible role of Ca2+-permeable TRPV1 in oxalate-mediated tubular cell damage and the depandance of such events of inflammation using both in LLC-PK1 cell line and in vivo in rat models. While the Authors presented the evidence for the relevance in the Introduction, it was not sufficiently mentioned some of the previous studies. Specifically,  studies involving crystal-induced inflammasome-mediated tissue injury, necroptosis signaling, and RIP kinase 3 in crystalline particle-induced cell necrosis (Sci Rep. 2017; 7: 15523). Furthermore, the pioneering work on the crystal-induced regulation by upstream store-operated Ca2+ entry signaling mechanism (Cell Death Discov. 2019; 5: 124) is also relevant since the present manuscript examined on the role TRPV1 Ca2+ channel in similar condition. Interestingly, above publication in Cell Death Discov. detailed with increased intracellular Ca2+ via STIM/ORAI and oxidative stress in the regulation of cell death and survival. However, the inflammatory pathway detailed in the present study can still give some new information.

They used both in vivo and in vitro model to produce supporting data. Scientific premise is strong; however, methodological approach is weak.   

Methodological concern:

In general, Bar diagrams in the Figures are not clearly shows the comparison between the groups. A bar can be added on the top the bars to state the significance between groups. It is important to mention in the Figure legends, which statistical test were employed to determine the differences between groups (unpaired t-test or one-way ANOVA). Some experiments used multiple group comparison, which means group/paired tests are necessary.

The mechanisms NLRP3 inflammasome activation is interesting, which shown to be activated by the increase in intracellular Ca2+. However, BAPTA blocked such pathway, and such process can activate store-operated Ca2+ entry possibly through STIM/ORAI, which can also cause cell damage. How the authors rule out those pathway activations needs to be justified.

LLC-PK1 is a tubular cell line derived from pig kidney, which may not be implicated in rat and humans, which show be discussed. Although the in vivo effects in rats are well done, some methodological illustrations would help the Readers.

In Figure 5D HP 14 days kidney IF picture is out of focus. Quality of IF pictures should be improved, some counterstained marker proteins might help.

What is the certainty that oxalate induced cell death involves caspase-1 dependent mechanism and not caspase 3 and 9. A variety of stimuli such as crystals, Ca2+ load, and reactive oxygen species (ROS) production in oxidative stress have been demonstrated to activate inflammasome? Moreover, intracellular Ca2+ rise can cause a number of detrimental pathway activation.

Did they measure H2O2 in the urine, it would be nice to see any in vivo effect.

Also measurements of the inflammatory paradigm IL-1/4/6 data would be helpful.

Author Response

Response to Reviewer 2’s comments

In this manuscript authors tested the possible role of Ca2+-permeable TRPV1 in oxalate-mediated tubular cell damage and the depandance of such events of inflammation using both in LLC-PK1 cell line and in vivo in rat models. While the Authors presented the evidence for the relevance in the Introduction, it was not sufficiently mentioned some of the previous studies. Specifically, studies involving crystal-induced inflammasome-mediated tissue injury, necroptosis signaling, and RIP kinase 3 in crystalline particle-induced cell necrosis (Sci Rep. 2017; 7: 15523). Furthermore, the pioneering work on the crystal-induced regulation by upstream store-operated Ca2+ entry signaling mechanism (Cell Death Discov. 2019; 5: 124) is also relevant since the present manuscript examined on the role TRPV1 Ca2+ channel in similar condition. Interestingly, above publication in Cell Death Discov. detailed with increased intracellular Ca2+ via STIM/ORAI and oxidative stress in the regulation of cell death and survival. However, the inflammatory pathway detailed in the present study can still give some new information.

Response: Thank you for your comments and provide valuable references to strengthen our background. We have amended these two references in the section of Introduction. Please see page 2, lines 77-82.

They used both in vivo and in vitro model to produce supporting data. Scientific premise is strong; however, methodological approach is weak.   

Methodological concern:

In general, Bar diagrams in the Figures are not clearly shows the comparison between the groups. A bar can be added on the top the bars to state the significance between groups. It is important to mention in the Figure legends, which statistical test were employed to determine the differences between groups (unpaired t-test or one-way ANOVA). Some experiments used multiple group comparison, which means group/paired tests are necessary.

Response: Thank you for your comments. We have added bars on the top the bars in each graph to show the significance between groups. The test method for statistic comparisons are also included in figure legends. We performed multiple group comparison by one-way ANOVA analysis among groups. Here we did not perform paired test. The density of fluo-3 in cells before treatment of calcium chelator (Figure 2A) and the level of chemiluminescence before treatment of lucigenin (Figure 6C) in renal cortical surface were only demonstrated to show a similar basal level in groups.    

The mechanisms NLRP3 inflammasome activation is interesting, which shown to be activated by the increase in intracellular Ca2+. However, BAPTA blocked such pathway, and such process can activate store-operated Ca2+ entry possibly through STIM/ORAI, which can also cause cell damage. How the authors rule out those pathway activations needs to be justified.

Response: Thank you for your comments. In this study, we showed that BAPTA attenuated calcium overload caused by oxalate. This treatment actually was associated with a decrease in oxalate-mediated LDH release (data not shown). This result is consistent with previous observations in a mixed (glia and neuron) cortical cultures showing that a non-toxic concentration (at microM) of BAPTA delays necrotic neuronal death caused by NMDA receptor overstimulation [Wie et al., 2001]. We therefore speculate that BAPTA treatment in this study probably maintains intracellular calcium at a near physiological concentration (Figure 2A) via an enhanced Stim/Orai-medaited store-operated calcium entry as per your suggestion because calcium is required for normal cell function for survival [Collatz et al., 1997; Zhang et al., 2018].

References

  1. Wie MB, Koh JY, Won MH, Lee JC, Shin TK, Moon CJ, Ha HJ, Park SM, Kim HC. BAPTA/AM, an intracellular calcium chelator, induces delayed necrosis by lipoxygenase-mediated free radicals in mouse cortical cultures. Prog Neuropsychopharmacol Biol Psychiatry. 2001 Nov;25(8):1641-59.
  2. Collatz MB, Rüdel R, Brinkmeier H. Intracellular calcium chelator BAPTA protects cells against toxic calcium overload but also alters physiological calcium responses. Cell Calcium. 1997 Jun;21(6):453-9.
  3. Zhang X, Gueguinou M, Trebak M. Store-Independent Orai Channels Regulated by STIM. In: Kozak JA, Putney JW Jr., editors. Calcium Entry Channels in Non-Excitable Cells. Boca Raton (FL): CRC Press/Taylor & Francis; 2018. Chapter 11. Available from: https://www.ncbi.nlm.nih.gov/books/NBK531427/ doi: 10.1201/9781315152592-11

LLC-PK1 is a tubular cell line derived from pig kidney, which may not be implicated in rat and humans, which show be discussed. Although the in vivo effects in rats are well done, some methodological illustrations would help the Readers.

Response: Thank you for your comments. LLC-PK1 were applied in this study because we previously showed that TRPV1 is expressed in this cell line [Lu et al., 2020]. In our pilot study, the protein expression of TRPV1 is more abundant than those in other human renal proximal tubule cells such as HK-2. Moreover, pig shares a number of surprising comparable traits with humans; for example, the porcine kidney is functionally and structurally similar to the human kidney. LLC-PK1 cells were therefore designed to explore the effects of oxalate. The data obtained from this cell line therefore may provide useful information for pre-clinical stage. We added an explanation as pig renal tubular epithelial kidney cells followed by the abbreviation of LLC-PK1 to clarify its source and to avoid confusion. Please see page 16, lines 496-497.

Reference

  1. Lu CL, Liao CH, Lu KC, Ma MC. TRPV1 Hyperfunction Involved in Uremic Toxin Indoxyl Sulfate-Mediated Renal Tubular Damage. Int J Mol Sci. 2020 Aug 27;21(17):6212. doi: 10.3390/ijms21176212. PMID: 32867359; PMCID: PMC7503230.

In Figure 5D HP 14 days kidney IF picture is out of focus. Quality of IF pictures should be improved, some counterstained marker proteins might help.

Response: Thank you for your comments. We tried to improve the quality of pictures especially those of the HP 14 days. Currently we do not have the data to show the co-localization of TRPV1 or AlOX12 with a specific marker of the proximal tubule as per your suggestion. We however cropped a local tissue in renal cortex as the inset picture (magnified at 400-fold) in Figure 5D to show the typical distribution of TRPV1 or ALOX12 in renal tubules. Please see the following picture or the revised Figure 5D.

Figure 5D. Representative pictures show tissue localization of TRPV1 (upper two pictures) and ALOX12 (lower two pictures) in renal cortex of control and 14-day HP kidneys at 200× magnification. Cell nuclei are counterstained with DAPI (blue). G, glomerulus. The local tissue of the renal cortex was magnified at 400× (insets) and demonstrated that TRPV1 and ALOX12 (green) are present at apical membrane (indicated by arrows) and cytosol (indicated by arrow heads) of renal tubules, respectively, in control and HP (14 days) kidneys. These expressions are increased in HP kidney.

What is the certainty that oxalate induced cell death involves caspase-1 dependent mechanism and not caspase 3 and 9. A variety of stimuli such as crystals, Ca2+ load, and reactive oxygen species (ROS) production in oxidative stress have been demonstrated to activate inflammasome? Moreover, intracellular Ca2+ rise can cause a number of detrimental pathway activation.

Response: Thank you for the comments. Previous studies in various cell or animal models showed that activation of different types of caspases involved in oxalate or calcium oxalate-induced tubular cell damage and inflammation [Mulay et al., 2015; Sun et al., 2020; Jeong et al., 2005; Song et al., 2020]. To determine caspases other than caspase-1 involved in oxalate nephropathy is also our interest. We agree that stimuli such as calcium oxalate crystals, calcium overload, and reactive oxygen species (ROS) production in oxidative stress were involved in activation of inflammasome. This study therefore focused on the effects of calcium and ROS because these two stimuli are related to TRPV1 function directly as previously reported [Zhai et al., 2020]. Moreover, the results of the present study showed that TRPV1 has a contributory role for calcium oxalate formation as damaged tubular cells or cell debris may act as a seed to enhance crystallization. We agree that intracellular calcium rise can activate a number of detrimental pathways. The present results in this study demonstrated one of detrimental pathways, inflammation, in cell damage caused by calcium overload.

References

  1. Mulay SR, Kulkarni OP, Rupanagudi KV, Migliorini A, Darisipudi MN, Vilaysane A, Muruve D, Shi Y, Munro F, Liapis H, Anders HJ. Calcium oxalate crystals induce renal inflammation by NLRP3-mediated IL-1β secretion. J Clin Invest. 2013 Jan;123(1):236-46. doi: 10.1172/JCI63679.
  2. Sun Y, Liu Y, Guan X, et al. Atorvastatin inhibits renal inflammatory response induced by calcium oxalate crystals via inhibiting the activation of TLR4/NF-κB and NLRP3 inflammasome. IUBMB Life. 2020;72: 1065–1074. https://doi.org/10.1002/iub.2250
  3. Jeong BC, Kwak C, Cho KS, Kim BS, Hong SK, Kim JI, Lee C, Kim HH. Apoptosis induced by oxalate in human renal tubular epithelial HK-2 cells. Urol Res. 2005 May;33(2):87-92. doi: 10.1007/s00240-004-0451-5.
  4. Song Q, He Z, Li B, Liu J, Liu L, Liao W, Xiong Y, Song C, Yang S, Liu Y. Melatonin inhibits oxalate-induced endoplasmic reticulum stress and apoptosis in HK-2 cells by activating the AMPK pathway. Cell Cycle. 2020 Oct;19(20):2600-2610. doi: 10.1080/15384101.2020.1810401.
  5. Lin, C.S.; Lee, S.H.; Huang, H.S.; Chen, Y.S.; Ma, M.C. H2O2 generated by NADPH oxidase 4 contributes to transient receptor potential vanilloid 1 channel-mediated mechanosensation in the rat kidney. Am J Physiol Renal Physiol. 2015 Aug 15;309(4):F369-76. doi: 10.1152/ajprenal.00462.2014.
  6. Zhai, K.; Liskova, A.; Kubatka, P.; Büsselberg, D. Calcium Entry through TRPV1: A Potential Target for the Regulation of Proliferation and Apoptosis in Cancerous and Healthy Cells. Int J Mol Sci. 2020 Jun 11;21(11):4177. doi: 10.3390/ijms21114177.

Did they measure H2O2 in the urine, it would be nice to see any in vivo effect.

Response: Thank you for your comments. Urinary levels of H2O2 were determined according to our previous study [Lin et al., 2015]. In the following Figure A, our results showed that urinary H2O2 levels are gradually increased after hyperoxaluric induction when compared to that in the control rat. A peak increase was found after 28 days of hyperoxaluria. Currently, we do not know whether increases in urinary H2O2 excretion are related to TRPV1 hyperfunction after hyperoxaluric induction. We will clarify this issue soon. Please allow us to leave the following data here because it is related to our future study.

Figure A. Effect of hyperoxaluria on urinary H2O2 excretion. Urine samples were collected daily from metabolic cage study to measure excreted H2O2 after hyperoxaluric (HP) induction. Urine in control (C) rats was collected only on day 28. Note that urinary H2O2 increases in a time-dependent manner in HP rats. *P < 0.05, HP vs. control.

References

  1. Lin CS, Lee SH, Huang HS, Chen YS, Ma MC. H2O2 generated by NADPH oxidase 4 contributes to transient receptor potential vanilloid 1 channel-mediated mechanosensation in the rat kidney. Am J Physiol Renal Physiol. 2015 Aug 15;309(4):F369-76. doi: 10.1152/ajprenal.00462.2014.

Also measurements of the inflammatory paradigm IL-1/4/6 data would be helpful.

Response: Thank you for your comments. Changes in IL-1β formation in tubular cells and in hyperoxaluric kidney were showed in Figures 4C, 4D, 6F, and 8E. These in vitro and in vivo data clearly demonstrated that inflammatory cytokine IL-1β is mediated by TRPV1 in oxalate nephropathy. We do not have data related to IL-4 and IL-6 now. Since these cytokines playing differential roles in pro- or anti-inflammation and also related to TRPV1 [Obi et al.; Zhang et al., 2017], they are also our interest of research in the future.

References

  1. Obi S, Nakajima T, Hasegawa T, Kikuchi H, Oguri G, Takahashi M, Nakamura F, Yamasoba T, Sakuma M, Toyoda S, Tei C, Inoue T. Heat induces interleukin-6 in skeletal muscle cells via TRPV1/PKC/CREB pathways. J Appl Physiol. 2017 Mar 1;122(3):683-694. doi: 10.1152/japplphysiol.00139.2016.
  2. Zhang J, Zhou Z, Zhang N, Jin W, Ren Y, Chen C. Establishment of preliminary regulatory network of TRPV1 and related cytokines. Saudi J Biol Sci. 2017 Mar;24(3):582-588. doi: 10.1016/j.sjbs.2017.01.029.

Round 2

Reviewer 1 Report

Manuscript has improved substantially and is significantly more clear.

While the authors added the bar graph comparisons on tops of the figures, I feel that can still be done in a more clarified way - perhaps just explained in the figure legends with the use and inclusion of appropriate symbols and descriptions of what they represent.

I would also change the overall conclusion that TRPV5 antagonism could play a potential role in the amelioration of calcium oxalate crystallization, as you cannot really make a conclusion about kidney stones in general.

Author Response

Response: Thank you for your comments. We appreciate very much for your time and effort on reviewing our work. We have revised Figures 1, 3, 4, 7, and 8 to use symbols and clearly show the statistical significance between groups. As per your suggestion, we also explain the difference indicated by the symbols in the figure legend.

We revised the last sentence in the section of Conclusion to describe that TRPV1 antagonism in this study is related to oxalate nephropathy instead of kidney stone. Please see page 16, line 499. Concerning TRPV5, previous studies have revealed that TRPV5 is quite different from other members of the vanilloid subfamily TRPV1-4. Unlike TRPV1, TRPV5 is not considered to be thermosensitive or ligand-activated [Vennekens et al.; Nilius et al., 2000]. Compared to other TRPV channels, TRPV5 is also very calcium-selective under physiological ionic condition [Bouron et al., 2015]. Moreover, loss-of-function of TRPV5 is known to induce severe hypercalciuria in calcium stone formation [van der Wijst et al., 2019]. We previously demonstrated that gain-of-function of TRPV1 is harmful to renal tubular cells [Lu et al., 2020]. These results clearly indicated that TRPV1 and TRPV5 play biological roles differed from each other in terms of regulation of tubular function. One cannot confuse TRPV1 with TRPV5.

References

  1. Vennekens R, Hoenderop JG, Prenen J, Stuiver M, Willems PH, Droogmans G, Nilius B, Bindels RJ. Permeation and gating properties of the novel epithelial Ca(2+) channel. J Biol Chem. 2000 Feb 11;275(6):3963-9. doi: 10.1074/jbc.275.6.3963.
  2. Nilius B, Vennekens R, Prenen J, Hoenderop JG, Bindels RJ, Droogmans G. Whole-cell and single channel monovalent cation currents through the novel rabbit epithelial Ca2+ channel ECaC. J Physiol. 2000 Sep 1;527 Pt 2(Pt 2):239-48. doi: 10.1111/j.1469-7793.2000.00239.x.
  3. Bouron A, Kiselyov K, Oberwinkler J. Permeation, regulation and control of expression of TRP channels by trace metal ions. Pflugers Arch. 2015 Jun;467(6):1143-64. doi: 10.1007/s00424-014-1590-3.
  4. van der Wijst J, van Goor MK, Schreuder MF, Hoenderop JG. TRPV5 in renal tubular calcium handling and its potential relevance for nephrolithiasis. Kidney Int. 2019 Dec;96(6):1283-1291.
  5. Lu CL, Liao CH, Lu KC, Ma MC. TRPV1 Hyperfunction Involved in Uremic Toxin Indoxyl Sulfate-Mediated Renal Tubular Damage. Int J Mol Sci. 2020 Aug 27;21(17):6212. doi: 10.3390/ijms21176212.

Reviewer 2 Report

Authors are responsive to my comments sufficiently. I recommend to publish this research.

Author Response

Response: Thank you for your comment. We appreciate very much for your time and effort on reviewing our work.